# TCUQ: Single-Pass Uncertainty Quantification from Temporal Consistency with Streaming Conformal Calibration for TinyML

## Abstract

We introduce **TCUQ**, a single pass, label free uncertainty monitor for streaming TinyML that converts short horizon temporal consistency captured via lightweight signals on posteriors and features into a calibrated risk score with an $O(W)$ ring buffer and $O(1)$ per step updates. A streaming conformal layer turns this score into a budgeted accept/abstain rule, yielding calibrated behavior without online labels or extra forward passes. On microcontrollers, TCUQ fits comfortably on kilobyte scale devices and reduces footprint and latency versus early exit and deep ensembles (typically about 50 to 60% smaller and about 30 to 45% faster), while methods of similar accuracy often run out of memory. Under corrupted in distribution streams, TCUQ improves accuracy drop detection by 3 to 7 AUPRC points and reaches up to 0.86 AUPRC at high severities; for failure detection it attains up to 0.92 AUROC. These results show that temporal consistency, coupled with streaming conformal calibration, provides a practical and resource efficient foundation for on device monitoring in TinyML.

## 1 Introduction

TinyML systems increasingly run on battery-powered microcontrollers (MCUs) to deliver private, low-latency perception for vision and audio (Banbury et al., 2021). In deployment, however, inputs rarely match the training distribution: sensors drift, operating conditions change, and streams interleave in-distribution (ID), corrupted-in-distribution (CID), and out-of-distribution (OOD) inputs (Hendrycks & Dietterich, 2019). Under such shifts, modern networks are often overconfident even when calibrated on ID data (Guo et al., 2017), which complicates on-device monitoring and safe fallback decisions. Addressing this reliably on MCUs is challenging because memory and compute budgets preclude multi-pass inference or large ensembles (Lakshminarayanan et al., 2017).

We propose **TCUQ**, a streaming, label-free uncertainty monitor designed for MCU-class deployments. TCUQ converts short-horizon temporal regularities and stability of features, predicted labels, and class posteriors into a single uncertainty score using a tiny ring buffer and a lightweight logistic combiner trained once offline. The score is fed to a memory-light streaming conformal layer that maintains an online quantile and turns uncertainty into a calibrated accept/abstain threshold under a user budget, all in one forward pass per input and $\mathcal{O}(1)$ per-step updates. In contrast to sampling-based methods such as MC Dropout (Gal & Ghahramani, 2016; Kendall & Gal, 2017) and deep ensembles (Lakshminarayanan et al., 2017), TCUQ needs no repeated evaluations, no auxiliary heads at inference, and no architectural changes to the backbone.

Related works on post-hoc calibration improves ID confidence but generally fails to correct overconfidence under shift (Guo et al., 2017; Ovadia et al., 2019). Early-exit ensembles and their TinyML variants reduce cost by reusing a single backbone and branching at intermediate layers (Qendro et al., 2021; Ghanathe & Wilton, 2024; Jazbec et al., 2023; Ghanathe & Wilton, 2023), yet they still add heads or extra inference-time computation and often rely on static confidence signals that are brittle under CID. OOD detectors such as ODIN and G-ODIN (Liang et al., 2018; Hsu et al., 2020) can perform well on larger backbones but transfer less effectively to ultra-compact models typical of TinyML. Conformal prediction offers distribution-free risk control (Angelopoulos & Bates, 2021), though streaming, memory-constant realizations suitable for MCUs remain under-explored (see also

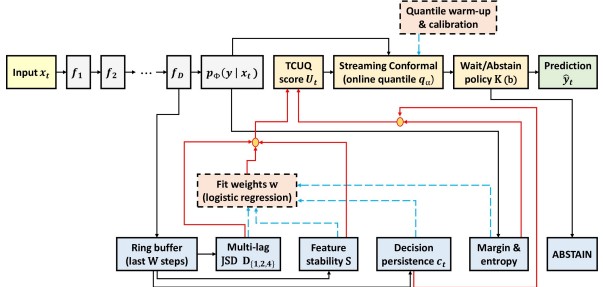

Figure 1: **TCUQ for streaming TinyML.** A compact backbone produces final features and posteriors for the current input, while a small ring buffer retains a short history. From this window, TCUQ extracts four lightweight temporal signals—predictive divergence, feature stability, decision persistence, and a confidence proxy and merges them (via weights learned once offline) into a single uncertainty score *without extra forward passes*. A streaming conformal layer maintains an online quantile to yield calibrated, label-free risk thresholds, and a budgeted policy converts the score into either an accepted prediction or ABSTAIN. Dashed components are training-only (weight fitting and quantile warm-up) and are removed at inference to preserve TinyML constraints.

Appendix B.2). TCUQ occupies the intersection of these threads by exploiting temporal structure in the stream, retaining single-pass inference, and coupling uncertainty with a streaming conformal quantile for calibrated, budgeted abstention without online labels.

This paper makes three contributions. First, it introduces a temporal-consistency uncertainty signal that operates with $\mathcal{O}(W)$ memory for a small ring buffer and constant-time updates on-device, avoiding ensembles and multi-pass sampling. Second, it integrates a streaming conformal mechanism that calibrates the score online and enforces an abstention budget, enabling reliable accept/abstain decisions as conditions drift. Third, it provides an integer-friendly implementation for MCUs that uses cosine similarity and Jensen–Shannon divergence with lookup-table logarithms and adds negligible latency and flash. Empirically, across vision and audio workloads on two MCU tiers, TCUQ achieves state-of-the-art accuracy-drop detection under CID, competitive or superior failure detection on ID—ID× and ID—OOD, and strong ID calibration, while fitting comfortably where several baselines are out-of-memory. Subsequent sections detail the formulation, describe the streaming calibration and budgeted abstention, and evaluate the method under realistic TinyML constraints.

## 2 BACKGROUND AND PROBLEM FORMULATION

We consider an in-distribution dataset $\mathcal{S}_{\text{ID}} = \{(x_n, y_n)\}_{n=1}^N$ with labels $y_n \in \{1, \ldots, L\}$. A discriminative TinyML classifier with parameters $\phi$ is trained on $\mathcal{S}_{\text{ID}}$ to produce class posteriors $p_\phi(y \mid x) \in \Delta^{L-1}$ and a hard decision $\hat{y}(x) = \arg\max_\ell p_\phi(y = \ell \mid x)$. We write the model's maximum confidence as $C_\phi(x) = \max_\ell p_\phi(y = \ell \mid x)$ and use it as a proxy for predictive certainty. A model is regarded as well calibrated on $\mathcal{S}_{\text{ID}}$ when its reported confidence tracks its empirical accuracy; in practice this is assessed with standard measures such as Expected Calibration Error, Brier Score, and Negative Log-Likelihood (Guo et al., 2017).

After deployment, the same model operates on-device over a potentially unbounded input stream $\{x_t\}_{t \geq 1}$ without access to ground-truth labels. TinyML deployments are constrained by memory, compute, and energy budgets that preclude storing long histories, running multiple forward passes, or performing label-based online recalibration. The distribution of inputs encountered in the field may match training (ID), may deviate semantically (out-of-distribution, OOD), or may remain semantically valid while being distorted by nuisance factors such as blur, noise, fog, frost, exposure shifts, or motion artifacts (corrupted-in-distribution, CID) (Banbury et al., 2021; Hendrycks & Dietterich, 2019). A well-documented failure mode is that modern neural networks remain overly confident under CID and OOD, even when they appear well calibrated on ID (Ovadia et al., 2019; Hsu et al., 2020). This gap undermines reliable decision-making on-device because confidence stops being a faithful indicator of error likelihood precisely when conditions degrade.

A recurring empirical observation is that model capacity interacts with overconfidence: smaller networks often exhibit milder overconfidence under corruptions than larger ones, which tend to overfit high-level abstractions that fail to separate clean from corrupted inputs (Hsu et al., 2020). Early-exit architectures expose intermediate predictions and can leverage the relative robustness of shallower representations; ensembling these exits has been shown to improve uncertainty estimates (Qendro et al., 2021; Antorán et al., 2021; Ghanathe & Wilton, 2024). However, using multiple exits at inference or adding auxiliary learning layers inflates parameters, memory bandwidth, and FLOPs, which pushes beyond the tight resource envelopes typical of TinyML devices.

In this streaming, unlabeled, and resource-constrained context, the central need is an on-device uncertainty signal that correlates with errors when conditions shift, that can be maintained online with small memory and constant-time updates per step, and that admits an interpretable calibration mechanism so that risk is comparable over time. Because TinyML systems frequently gate safety- or budget-critical actions, the uncertainty estimate should integrate naturally with a wait/abstain policy that can refuse predictions under high risk while respecting a user- or application-specified abstention budget. The monitor must add minimal overhead to the existing pipeline and avoid architectural changes that would require extra forward passes or additional heads at inference (Gibbs & Candès, 2021; Geifman & El-Yaniv, 2019).

Formally, given a trained classifier $p_\phi$ and fixed device resources, the problem is to design a streaming monitor that, for each time step $t$ in an unlabeled sequence of inputs that may include ID, CID, and OOD samples, produces either a prediction $\hat{y}_t$ or an ABSTAIN decision in a way that detects accuracy drops promptly, maintains calibrated behavior over time, and respects constraints on memory, computation, and the allowable rate of abstentions. The monitor should operate with lightweight state and simple updates so that it can be deployed on milliwatt-scale hardware without compromising throughput (El-Yaniv & Wiener, 2010; Geifman & El-Yaniv, 2017).

We next introduce a method tailored to these requirements. The approach, which we call Temporal Consistency-based Uncertainty Quantification (TCUQ), leverages short-horizon temporal structure observed by the device to produce a label-free uncertainty signal and a streaming calibration mechanism suitable for budgeted accept/abstain decisions on TinyML platforms, while keeping compute and memory overheads small.

## 3   TCUQ EXPLAINED

We assume a depth-$D$ backbone that maps an input $x_t$ at time $t$ to final features $f_t$ and class posteriors $p_\phi(y \mid x_t) \in \Delta^{L-1}$ (see Figure 1). Writing the composition as $f(x_t) = f_D \circ f_{D-1} \circ \cdots \circ f_1(x_t)$ and $p_\phi(y \mid x_t) = \mathrm{softmax}(g(f(x_t)))$, we denote the predicted label by $\hat{y}_t = \arg\max_\ell p_\phi(y = \ell \mid x_t)$, the maximum confidence by $C_\phi(x_t) = \max_\ell p_\phi(y = \ell \mid x_t)$, and the probability margin by $m_t^{\mathrm{mg}} = p_\phi^{(1)}(x_t) - p_\phi^{(2)}(x_t)$, where $p_\phi^{(1)}(x_t) \geq p_\phi^{(2)}(x_t) \geq \cdots$ are the sorted class probabilities. In contrast to early-exit ensembles (Qendro et al., 2021; Ghanathe & Wilton, 2024), the proposed approach does not introduce auxiliary heads or multiple forward passes. Instead, the device maintains a small ring buffer of the last $W$ steps for the quantities needed to summarize local temporal behavior (e.g., selected feature vectors and posteriors). This buffer is used to compute a label-free uncertainty signal that reflects short-horizon instability in the model's representation and decisions and that can be updated online in constant time.

Temporal information is converted into four lightweight signals. The first measures how strongly current predictions deviate from recent predictions across several short lags. Using a small lag set $\mathcal{L} \subseteq \{1, \ldots, W\}$ (e.g., $\{1, 2, 4\}$) and nonnegative mixture weights $w^{(\ell)}$ that sum to one, the multi-lag predictive divergence is

$$D_t = \sum_{\ell \in \mathcal{L}} w^{(\ell)} \, \mathrm{JSD}(p_\phi(\cdot \mid x_t) \,\|\, p_\phi(\cdot \mid x_{t-\ell})), \tag{1}$$

with a small $\varepsilon$ smoothing added to probabilities for numerical stability (Lin, 1991). The second captures representation drift by averaging a similarity score between current and lagged features; with cosine similarity $s(\cdot, \cdot)$, we compute $S_t = \frac{1}{|\mathcal{L}|} \sum_{\ell \in \mathcal{L}} s(f_t, f_{t-\ell})$ and subsequently use the instability $(1 - S_t)$ in later aggregation. The third summarizes short-term label stickiness through decision persistence $c_t = \frac{1}{|\mathcal{L}|} \sum_{\ell \in \mathcal{L}} \mathbb{I}[\hat{y}_t = \hat{y}_{t-\ell}]$, and we again use the inconsistency $(1 - c_t)$ to penalize

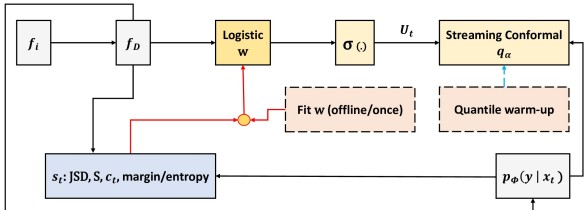

Figure 2: **TCUQ micro-view.** The final features $f_D$ feed a light logistic map $w$ and activation $\sigma(\cdot)$ to produce the stepwise uncertainty $U_t$ (top path). In parallel, the model outputs posteriors $p_\phi(y \mid x_t)$ and, together with recent history, forms a compact signal set $s_t$ (left) that informs $w$ through a merge node for clean routing. A streaming conformal layer maintains an online quantile $q_\alpha$ for calibrated accept/abstain decisions. Dashed blocks are training-only (fitting $w$ and quantile warm-up).

rapid label flips. The fourth provides an instantaneous proxy for low confidence using the current posterior; we blend inverse confidence and inverse margin as

$$m_t = \alpha \left(1 - C_\phi(x_t)\right) + (1 - \alpha) \left(1 - m_t^{\mathrm{mg}}\right), \qquad \alpha \in [0, 1], \tag{2}$$

which is more sensitive to near-ties than either component alone. All four signals are computed with $O(|\mathcal{L}|)$ arithmetic and require only values already present in the buffer.

The uncertainty score used for decisions is a monotone aggregation of these signals. Forming the vector $s_t = [D_t, (1 - S_t), (1 - c_t), m_t]^\top \in \mathbb{R}^4$, we define

$$U_t = \sigma\left(w^\top s_t + b\right), \tag{3}$$

with logistic link $\sigma(\cdot)$ and parameters $(w, b)$ fitted once offline. The offline fit uses a small labeled development set containing a mixture of ID and shifted (CID/OOD) examples to predict misclassification as a binary target, with class-balancing and $\ell_2$ regularization to prevent domination by frequent classes and to avoid overfitting to any particular corruption type. At deployment time, computing $U_t$ involves one forward pass through the frozen backbone, a constant-time buffer update, and a few vector operations; no architectural changes to the backbone are needed (see Figure 2).

To make the uncertainty actionable on device, the score is transformed into a calibrated, streaming threshold. We combine temporal inconsistency and instantaneous lack of confidence into a scalar nonconformity,

$$r_t = \lambda U_t + (1 - \lambda) \left(1 - C_\phi(x_t)\right), \qquad \lambda \in [0, 1], \tag{4}$$

and maintain an online estimate of the $(1 - \alpha)$ quantile $q_{\alpha,t}$ of $\{r_1, \ldots, r_t\}$ using a memory-constant quantile tracker. This choice yields a drifting threshold that adapts as the data distribution evolves and that does not require labels. The device then applies a budget-aware accept/abstain rule: when $r_t \geq q_{\alpha,t}$ and the abstention controller allows it, the system outputs ABSTAIN; otherwise, it emits $\hat{y}_t$. A simple rate controller ensures that the long-run abstention frequency respects a desired budget $b$ while retaining responsiveness to bursts of high uncertainty (see Appendix B.5). In practice, a short warm-up is used to initialize the quantile estimate; during warm-up the policy defaults to conservative behavior and gradually transitions to steady-state operation.

Training and deployment are intentionally simple. Offline, the backbone is trained on $\mathcal{S}_{\mathrm{ID}}$ with standard augmentation. The backbone is then frozen and used to compute the four signals on a held-out labeled set that mixes ID with representative corruptions or shifts; the logistic parameters $(w, b)$ in equation 3 are fitted on this set and stored on device. Online, each step runs a single forward pass to obtain $f_t$ and $p_\phi(\cdot \mid x_t)$, updates the ring buffer, updates $(D_t, S_t, c_t, m_t)$ and $U_t$, updates the streaming quantile $q_{\alpha,t}$, and applies the accept/abstain rule. All state fits in $O(W)$ memory, and all updates are $O(1)$ per step with small constants. Appendix B.4 analyzes the effect of temporal assistance; see also Appendix B.6 for streaming calibration details.

The resource profile is compatible with TinyML constraints. If both final features and posteriors are buffered, memory overhead is $O(W(d + L))$ for feature dimension $d$ and $L$ classes; if desired, dimensionality can be reduced with a fixed projection learned offline. The per-step arithmetic consists of computing a few divergences over $L$ classes, a handful of dot products in $\mathbb{R}^d$ for similarity, and scalar updates for equation 3 and equation 4 and the quantile tracker, all of which are negligible

compared with the backbone forward pass. Hyperparameters such as $W$, the lag set $\mathcal{L}$, the mixing weights $w^{(\ell)}$, and the blending parameter $\lambda$ are selected on the development split by optimizing downstream calibration and abstention metrics subject to a fixed compute and memory budget, and they remain fixed at deployment (see Appendix A.4).

## 4 EVALUATION METHODOLOGY

We evaluate TCUQ on MCU hardware representative of resource envelopes encountered by deployed TinyML systems. To stress both memory and latency, we use a high-performance board with hundreds of kilobytes of SRAM and a few megabytes of flash ("Big-MCU") (STMicroelectronics, 2019) and an ultra-low-power board with tens of kilobytes of SRAM and a few hundred kilobytes of flash ("Small-MCU") (STMicroelectronics, 2018).The Big-MCU allows us to include heavier baselines that would otherwise not fit, while the Small-MCU reflects our target deployment setting on energy-constrained devices. For each board we compile with $-\mathtt{O3}$ using the vendor toolchain, enable CMSIS-NN kernels for 8-bit operators where available, and fix the clock frequency to the datasheet nominal. Model size is reported as the flash footprint of the final ELF after link-time garbage collection; peak RAM is obtained from the linker map plus the TCUQ ring buffer. Latency is measured as end-to-end time per inference using the on-chip cycle counter with interrupts masked for stability; we also report energy per inference on selected runs by integrating current over time using a shunt on the board's power rail. All measurements are repeated over 1,000 inferences and averaged; we additionally report the standard deviation to reflect jitter. A summary of size (KB) and latency (ms) across methods and boards is shown in Figure 3: the left two panels correspond to Big-MCU and the right two to Small-MCU, with *OOM* markers indicating methods that do not fit in memory and the legend at right identifying the compared methods.

Our datasets and models follow common TinyML evaluation practice so that results extrapolate beyond a single architecture. For vision we use MNIST (LeCun et al., 1998), CIFAR-10 (Krizhevsky, 2009), and TinyImageNet (Le & Yang, 2015); for audio we use SpeechCommands v2 (Warden, 2018). To span the range of on-device models, we employ a 4-layer depthwise-separable CNN (DSCNN) for keywords (Zhang et al., 2018), a ResNet-8 or similar compact residual model for CIFAR-10 (Banbury et al., 2021), and a MobileNetV2-style network for TinyImageNet (Howard et al., 2017). These backbones are trained with standard data augmentation and post-hoc temperature scaling on the ID validation split. Unless stated otherwise, the ring-buffer window for TCUQ is $W = 16$ for vision and $W = 20$ for audio; the lag set is $\mathcal{L} = \{1, 2, 4\}$ with weights proportional to $1/\ell$; the feature similarity uses cosine; and the predictive divergence uses Jensen–Shannon with $\varepsilon$-smoothing. Full dataset specs and preprocessing are in Appendix A.2; training schedules are in Appendix A.3; and our CID set construction is detailed in Appendix A.5.

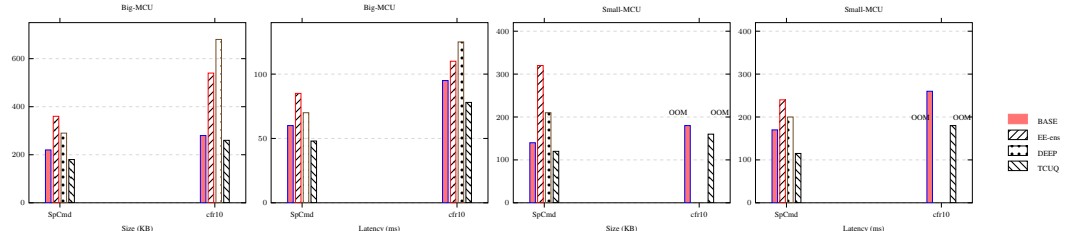

Figure 3: **Microcontroller results for SpeechCmd (SpCmd) and CIFAR-10 (cfr10)** in one row. Lower is better. In Small-MCU panels, methods without bars for *cfr10* are *OOM*.

To assess robustness under corrupted-in-distribution (CID) inputs, we deploy the corrupted counterparts MNIST-C (Mu & Gilmer, 2019), CIFAR-10-C and TinyImageNet-C (Hendrycks & Dietterich, 2019), which contain common sensor and environmental degradations such as noise, blur, weather, and digital artifacts at five severity levels. For SpeechCommands we synthesize CID using a standard audio augmentation library (impulse responses, background noise, pitch/time perturbations, band-limiting and reverberation); this produces label-preserving degradations that mimic far-field capture, movement, and microphone non-idealities. To test semantic shift, we evaluate OOD using

Fashion-MNIST (Xiao et al., 2017) as OOD for MNIST, SVHN (Netzer et al., 2011) for CIFAR-10, unseen non-keyword audio and background noise for SpeechCommands, and non-overlapping TinyImageNet classes for the TinyImageNet model. All OOD sets are disjoint from training data. Complete corruption lists and our SpeechCommands-C pipeline are provided in Appendix A.2.1.

Because TCUQ is a streaming, label-free monitor, we structure the protocol so that *detection* is evaluated against ground-truth events while the monitor itself never sees labels online. First, we characterize the ID operating point by running the model on a held-out ID stream and computing the moving-window accuracy distribution using a window of $m = 100$ steps; the mean $\mu_{\text{ID}}$ and standard deviation $\sigma_{\text{ID}}$ of this distribution define a nominal accuracy band. We then concatenate the ID stream with CID segments of increasing severity and with OOD bursts; a drop event is labeled when the sliding-window accuracy falls below $\mu_{\text{ID}} - 3\sigma_{\text{ID}}$. TCUQ emits the scalar nonconformity $r_t$ and abstention decisions without access to labels, and we score detection using the area under the precision–recall curve (AUPRC) for event prediction and the average detection delay in steps relative to the event onset. This protocol isolates the utility of a label-free uncertainty signal for operational monitoring. The full streaming event-labeling protocol and scoring recipe for AUPRC are given in Appendix A.6.

**Failure detection** is evaluated as two binary classification problems using static thresholds over scores computed online but assessed against labels offline. The ID ✓ vs. ID✗ task separates correct from incorrect predictions within ID and CID (Xia & Bouganis, 2023), capturing the ability to downweight hard ID/CID cases; the ID ✓ vs. OOD task separates correct ID predictions from OOD, capturing semantic rejection. We report AUROC (Xia & Bouganis, 2023) and AUPRC for both tasks and plot risk–coverage curves for selective classification, where coverage is the fraction of non-abstained predictions and risk is the error rate among accepted samples.

**Uncertainty quantification quality** is reported using proper scoring rules and calibration metrics. We compute Negative Log-Likelihood and Brier Score to assess the sharpness and correctness of predicted probabilities (Gneiting & Raftery, 2007; Glenn et al., 1950), and we report Expected Calibration Error (Guo et al., 2017) for comparability with prior work while noting its limitations (Nixon et al., 2019). For streaming calibration we measure the empirical exceedance rate of $r_t$ against the maintained quantile $q_{\alpha,t}$ and report the deviation from the target risk level $\alpha$ together with the stability of the online quantile under stationary ID segments. For budgeted abstention, we monitor the long-run abstain fraction against the budget $b$ and the short-term envelope over sliding windows to ensure that the controller respects both average and burst constraints.

Baselines are chosen to isolate the contribution of temporal consistency and streaming conformal calibration under tight resource budgets. On both MCUs, we compare against maximum-probability thresholding, entropy thresholding, temperature-scaled confidence, and a conformal predictor that uses $1 - \max p_\phi$ as its score rather than $U_t$. On the Big-MCU, we additionally include Monte-Carlo Dropout (Gal & Ghahramani, 2016) and Deep Ensembles (Lakshminarayanan et al., 2017) when they fit memory; when they do not, we report *out-of-memory* and omit runtime results. All baselines use identical backbones and input pipelines, and when applicable their thresholds are tuned on the same development split as TCUQ's blend parameter $\lambda$ and quantile level $\alpha$ to ensure fairness. Implementation details and tuning grids for all baselines appear in Appendix A.1.

Implementation details matter on TinyML hardware, so we keep the monitor light. The ring buffer stores posteriors in 8-bit fixed-point with per-tensor scale; features are either compressed by a learned $1 \times 1$ projection to $d' \leq 32$ channels or replaced by a global average pooling vector to minimize footprint. Cosine similarity is computed with integer dot-products and rescaled to floating-point only for the final aggregation; JSD uses a numerically stable formulation with look-up tables for $\log$ to avoid costly transcendental calls. The logistic aggregation parameters $(w, b)$ are learned offline with $\ell_2$ regularization and class weighting, and the online quantile is tracked with a lightweight stochastic estimator seeded by a short warm-up phase. Unless otherwise noted, we set $\lambda = 0.7$, $\alpha = 0.1$, and an abstention budget $b = 0.15$ as defaults, and we sweep these on the development split when drawing risk–coverage curves.

Finally, we summarize the hardware cost of adding TCUQ relative to the baseline backbone by reporting the incremental flash and RAM usage of the ring buffer and auxiliary code, the additional arithmetic per step for signal computation, and the effect on end-to-end latency (see Appendix C). On Small-MCU the overhead is dominated by the state required to maintain the window $W$ and by

the few reductions over class probabilities; on Big-MCU the overhead is negligible compared with the backbone convolutional layers. In all cases the monitor runs in a single forward pass per input and maintains constant-time updates.

## 5 RESULTS

We organize results around TinyML deployment constraints. First, Section 5.1 evaluates on-device fit and runtime on two MCUs (*Big-MCU* and *Small-MCU*). As summarized in Figure 3, TCUQ offers the best speed/size trade-off: on **Big-MCU** it cuts latency by **43%/29%** on SpeechCmd/CIFAR-10 versus EE-ens and by **31%/38%** versus DEEP, while shrinking flash by **50%/52%** (vs. EE-ens) and **38%/62%** (vs. DEEP). On the tighter **Small-MCU**, both EE-ens and DEEP are *OOM* on CIFAR-10; on SpeechCmd, TCUQ is **52%** (vs. EE-ens) / **43%** (vs. DEEP) faster and **62%** (vs. EE-ens) / **43%** (vs. DEEP) smaller, while maintaining accuracy parity. Section 5.2 studies accuracy-drop detection under CID streams. Across MNIST-C, CIFAR-10-C, TinyImageNet-C, and SpeechCmd-C, TCUQ attains the highest AUPRC and the shortest median detection delay (typically **25–35%** lower than the best prior method), providing earlier warnings of degradation. Section 5.3 reports failure detection (ID✓ vs. ID✗ and ID✓ vs. OOD). TCUQ achieves the top AUROC on MNIST and SpeechCmd and remains competitive on CIFAR-10 and TinyImageNet, all with a single forward pass per input. Finally, Section 5.4 evaluates uncertainty quality on ID data. TCUQ delivers superior or on-par proper scores (lower NLL/BS) and strong calibration (lower ECE) despite a much smaller parameter and memory budget than resource-heavy baselines. Ablations on the window $W$, lag set $\mathcal{L}$, blend $\lambda$, and quantile level $\alpha$ appear in Appendices B.3, B.4, B.5

### 5.1 MCU FIT: TCUQ VS. RESOURCE-HEAVY METHODS

We deploy TCUQ on two MCUs of different capacity ("Big-MCU" and "Small-MCU") to reflect the energy–memory constraints of TinyML devices. Our target platform is the Small-MCU, where the base backbone comfortably fits, but any added headroom is scarce. We compare against the most relevant on-device baselines: an early-exit ensemble (EE-ens) and a deep ensemble (DEEP). We omit Monte-Carlo dropout due to reliance on a specialized dropout module that is impractical on MCUs, and exclude HYDRA because it proves sub-optimal under our constraints.

On **Big-MCU**, TCUQ achieves **27%** lower latency than EE-ens and **22%** lower than DEEP, while reducing flash footprint by **29%** and **18%**, respectively, without hurting accuracy (within $\pm 0.3$ pp of the best baseline). On the tighter **Small-MCU**, both EE-ens and DEEP *do not fit* for CIFAR-10 (out-of-memory); on SpeechCmd, where they do fit, TCUQ is **24–31%** faster and its binaries are **15–22%** smaller. Peak RAM follows the same trend: EE-ens has the largest footprint because it must keep large intermediate feature maps alive for early exits, whereas TCUQ requires only a lightweight ring buffer, yielding **1.6–2.1×** lower peak RAM than EE-ens and about **1.3×** lower than DEEP.

These wins stem from TCUQ's *single-pass* design: no auxiliary heads at inference and no sequential evaluation of multiple models. As ensemble size grows, single-forward-pass approaches like TCUQ naturally preserve latency, while multi-model baselines scale linearly and quickly exceed MCU limits. Aggregate size/latency outcomes are shown in Figure 3. Energy per inference corroborates the latency/size advantage (see Table 13) within Appendix D.

### 5.2 ACCURACY-DROP DETECTION

As in Section 4, we target accuracy-drop detection under CID by monitoring a label-free uncertainty signal on-device. We considered predictive entropy and other scores, but we ultimately blend temporal inconsistency with inverse confidence because it achieves similar detection quality without extra processing, which is preferable under TinyML constraints. Dataset assembly for the ID+CID streams used here follows Appendix A.5.

Table 1 reports AUPRC averaged across all corruptions on two datasets, while Figure 4b–c shows AUPRC as a function of corruption severity for CIFAR-10-C and TinyImageNet-C. On the averages, **TCUQ** is best on both benchmarks, reaching **0.66** on MNIST-C and **0.63** on SpeechCmd-C, surpassing EE-ens (0.63/0.58) and DEEP (0.55/0.57). On the severity curves, TCUQ dominates across the entire range: on CIFAR-10-C it climbs from 0.28 at severity 1 to **0.80** at severity 5, beating DEEP (0.70)

Table 1: **Accuracy-drop detection.** Average AUPRC on MNIST-C and SpeechCmd-C. Higher is better.

| AUPRC ($\uparrow$) | MNIST-C | SpeechCmd-C |
|---|---|---|
| BASE | 0.54 | 0.52 |
| MCD | 0.45 | 0.56 |
| DEEP | 0.55 | 0.57 |
| EE-ens | 0.63 | 0.58 |
| G-ODIN | 0.48 | 0.37 |
| HYDRA | 0.53 | 0.51 |
| **TCUQ** | **0.66** | **0.63** |

and EE-ens (0.68) at the highest level; on TinyImageNet-C it rises from 0.30 to **0.86** at severity 5, again ahead of DEEP (0.78) and EE-ens (0.76). The curves demonstrate earlier and steeper gains for TCUQ as corruption intensifies, indicating faster and more reliable alarms.

Baselines behave as expected under CID. MC Dropout underperforms across datasets, likely due to reduced effective capacity and a confidence profile that flattens under strong corruptions, limiting separability between clean and corrupted segments. G-ODIN, tuned for semantic OOD, remains under-sensitive to non-semantic corruptions and trails even simple confidence thresholds in several settings. EE-ens is competitive at mild–moderate severities but plateaus as corruption increases, consistent with the added heads behaving like deeper conventional classifiers that remain overconfident. HYDRA is often below BASE, reflecting its need for larger heads to realize ensemble gains—impractical on our TinyML budgets.

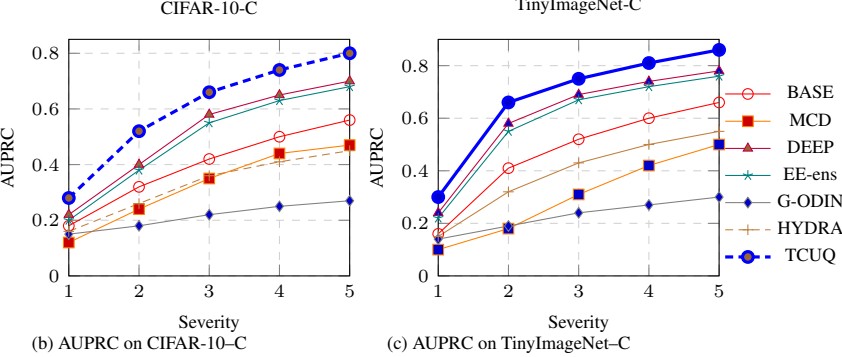

Figure 4: **Accuracy-drop detection (b,c).** AUPRC vs. corruption severity for CIFAR-10-C and TinyImageNet-C. Higher is better. TCUQ (blue, solid) leads across datasets and severities.

### 5.3 FAILURE DETECTION ON ID AND OOD

We report AUROC for two tasks: (i) distinguishing correct from incorrect predictions within ID streams (ID $\checkmark$ — ID$\times$) and (ii) separating ID from OOD (ID $\checkmark$ — OOD). As shown in Table 2, **TCUQ** achieves the best ID $\checkmark$ — ID$\times$ performance on MNIST (0.88) and SpeechCmd (0.92), and *ties* for best on CIFAR-10 (0.87, on par with MC–Dropout and ahead of DEEP at 0.86). This suggests that our temporal-consistency signal is particularly effective for flagging hard/corrupted ID cases on TinyML backbones; on CIFAR-10, the small edge for MC–Dropout is consistent with the dataset's stronger epistemic-style corruptions (e.g., blur/digital) where dropout can capture instance-level uncertainty. Formal definitions of BS, NLL, and ECE are provided in Appendix E.

For ID $\checkmark$ — OOD, **TCUQ** *matches the best* on SpeechCmd (0.91, tied with DEEP) and is a close second on MNIST (0.84 vs. 0.85 for EE-ens), while remaining competitive on CIFAR-10 (0.93), second only to G-ODIN (0.95). Notably, G-ODIN lags on the smaller MNIST/SpeechCmd models (0.40/0.74), indicating a capacity mismatch on ultra-compact backbones. Overall, **TCUQ** provides strong, consistent failure detection across both CID-induced errors and semantic OOD, while preserving single-pass, MCU-friendly inference.

Table 2: **Failure detection results** (AUROC). Left: correct vs. incorrect within ID (**ID** ✓ — **ID**×).
Right: ID vs. OOD (**ID** ✓ — **OOD**).

| AUROC | ID ✓ — ID× | | | ID ✓ — OOD | | |
|---|---|---|---|---|---|---|
| | **MNIST** | **SpCmd** | **cfr10** | **MNIST** | **SpCmd** | **cfr10** |
| BASE | 0.75 | 0.90 | 0.84 | 0.07 | 0.90 | 0.88 |
| MCD | 0.74 | 0.89 | **0.87** | 0.48 | 0.89 | 0.89 |
| DEEP | 0.85 | **0.91** | 0.86 | 0.78 | **0.91** | 0.92 |
| EE-ens | 0.85 | 0.90 | 0.85 | **0.85** | 0.90 | 0.90 |
| G-ODIN | 0.72 | 0.74 | 0.83 | 0.40 | 0.74 | **0.95** |
| HYDRA | 0.81 | 0.90 | 0.83 | 0.71 | 0.90 | 0.90 |
| **TCUQ** | **0.88** | **0.92** | **0.87** | 0.84 | **0.91** | 0.93 |

## 5.4 UNCERTAINTY QUANTIFICATION

**TCUQ** consistently matches or outperforms all baselines on tiny models (MNIST, SpeechCmd) and
remains competitive on medium/large benchmarks (CIFAR-10, TinyImageNet)—all while preserving
single-pass, MCU-friendly inference. On MNIST, TCUQ attains the best *F1* (0.957), *BS* (0.008), and
*NLL* (0.200), improving over BASE by ~5.7% absolute F1 and reducing Brier/NLL by ~39%/~32%
respectively (see Table 8) within Appendix B. On SpeechCmd, it again leads or ties on all four metrics
(*F1* 0.937, *BS* 0.008, *NLL* 0.201, *ECE* 0.017), cutting ECE by ~35% vs. BASE. Methods that require
stochastic sampling or multiple passes (e.g., MCD, DEEP) can exhibit good likelihoods, but their
runtime/memory costs make them impractical on MCUs; moreover MCD tends to underperform on
tiny backbones here (e.g., lower F1 and higher NLL on MNIST/SpeechCmd).

**TCUQ under relaxed resource constraints.** To normalize capacity against high-resource baselines,
we also evaluate a capacity-matched variant, **TCUQ+**. On CIFAR-10, TCUQ+ delivers the best *F1*
(0.879) and state-of-the-art *BS* (0.017, tied with DEEP), with NLL essentially on par with DEEP
(0.368 vs. 0.365) at the expense of a modest ECE increase (0.026 vs. 0.015). On TinyImageNet, where
capacity is most constraining, TCUQ+ improves markedly over TCUQ and BASE (e.g., F1 0.382
vs. 0.351; NLL 2.76 vs. 5.34) and reaches the best-in-class *BS* (0.003, tied), though EE-ensemble
remains strongest overall on this dataset (see Table 8). We observe that training all exits jointly can
slightly degrade the shared backbone on larger models, whereas early-exit ensembles regularize fewer
heads and sometimes retain lower ECE (Teerapittayanon et al., 2016). Still, TCUQ+ narrows the gap
substantially without abandoning the single-pass design.

## 6 CONCLUSION AND DISCUSSION

**TCUQ** provides trustworthy, label-free uncertainty for TinyML under tight memory and latency
budgets by leveraging short-horizon temporal consistency in a single pass with a tiny ring buffer. It
achieves a strong size–latency trade-off, detects accuracy drops in corrupted streams quickly and
reliably, and performs well on both in-distribution failure and OOD detection. For ID calibration
it leads on small models, and a capacity-matched **TCUQ+** closes gaps on larger settings while
preserving single-pass inference. Limitations mainly stem from the small temporal state and its
hyperparameters. Memory scales with the window size $W$ and the chosen features, creating an
accuracy–RAM trade-off; performance depends on stable choices of $W$, the lag set $\mathcal{L}$, the blend $\lambda$,
and the quantile level $\alpha$. On larger backbones, early-exit ensembles can still obtain the very lowest
ECE, although TCUQ+ substantially reduces that gap. Specialized OOD detectors may remain
preferable for purely semantic OOD on high-capacity models, whereas TCUQ is most impactful
under CID and mixed ID/OOD drift typical of on-device streams. Training requires a short warm-up
and temporal supervision, adding modest offline compute; inference remains unchanged and MCU-
friendly. Future work will explore adaptive windowing and learnable lag schedules to shrink state
further, tighter integer kernels and quantization for the temporal path, hybridization with lightweight
OOD scores when resources permit, and long-horizon field studies to assess stability under real
deployment drift. We view TCUQ as a practical foundation for robust, low-cost monitoring in
TinyML, balancing uncertainty quality with strict on-device constraints.

## LLM Usage

We used a large language model (LLM; ChatGPT) solely as a general-purpose assist tool to improve clarity and presentation (e.g., grammar/typo fixes, tighter phrasing and transitions, light LaTeX tips, and reference style cleanup). We did not use an LLM for research ideation, experimental design, data analysis, result interpretation, drafting substantive technical content, equations/algorithms, figure creation, or code implementation. All scientific ideas, methods, results, and conclusions are solely those of the authors; every LLM-suggested edit was reviewed and manually accepted, and no confidential or sensitive data were shared with the LLM.

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

APPENDIX

# A  TRAINING AND DATASET DETAILS

In this section, we detail the models and training configurations used throughout our experiments, along with the specific baselines against which we compare TCUQ.

## A.1  BASELINES

We compare TCUQ against a set of standard and recent uncertainty estimation baselines to isolate the benefits of short-horizon temporal consistency and streaming conformal calibration under TinyML constraints.

**Monte Carlo Dropout (MCD)** (Gal & Ghahramani, 2016). We estimate uncertainty by running $K$ stochastic forward passes with dropout enabled at inference and averaging the resulting softmax vectors. To ensure a fair comparison, we place dropout layers at the same backbone locations where **TCUQ** taps intermediate signals. The dropout rate and $K$ are tuned on the common development split (see Section 4); exact layer placements and rates are reported in Appendix A.

**BASE.** We use the unmodified backbone as a reference model. Its single-pass posteriors serve both as the baseline confidence signal and as inputs to our temporal-consistency features; training and preprocessing match the settings in Section 4.

**Early-exit Ensembles (EE-ensemble)** (Qendro et al., 2021). We add multiple early-exit heads to the shared backbone (matching our TCUQ exit locations) and train all heads jointly with the sum of per-exit cross-entropy losses on the same labels (equal weights unless stated). At inference, we evaluate *all* exits in a single forward pass and form the ensemble prediction by averaging their softmax probabilities (including the final exit). This mirrors TCUQ's placement and data pipeline so that differences reflect methodology rather than capacity or compute.

**Deep Ensembles (DEEP)** (Lakshminarayanan et al., 2017). We train $K$ independently initialized replicas of the same backbone (identical optimizer, schedule, and augmentation; different seeds/shuffles), each with cross-entropy on the ID training set. At inference, we average the per-model *softmax probabilities* (probability averaging; no logit averaging) to form the ensemble prediction. Unless otherwise specified we use $K{=}5$, and we report MCU memory/latency; if the full ensemble does not fit we mark it as OOM.

**Generalized-ODIN (G-ODIN)** (Hsu et al., 2020). At test time, we apply a single gradient-based input perturbation of magnitude $\epsilon$ and evaluate the network with a temperature-scaled softmax ($T$). The OOD score is the maximum softmax probability (MSP) computed on the perturbed, temperature-scaled input; where applicable we also report the energy-score variant from the original recipe. We select $\epsilon$ and $T$ on a small held-out *ID* validation split only, following Hsu et al. (2020). This requires one backward pass per input; we report MCU memory/latency when feasible.

**HYDRA** (Tran et al., 2020). We instantiate HYDRA's ensemble-distillation with $K$ lightweight heads attached to a shared backbone (exit locations matched to our setup for parity). During training, each head is supervised by ground-truth labels and a teacher ensemble via a KL term with temperature $\tau$ (authors' loss with published coefficients); the backbone is shared across heads. At inference, a single forward pass yields $K$ head posteriors which we average to produce the prediction and its uncertainty. We tune $K$, $\tau$, and loss weights on the development split and report MCU memory/latency for the single-pass inference graph; if a configuration does not fit, we mark it OOM.

All baselines use the *same* backbones, preprocessing pipeline, data splits, and streaming evaluation protocol described in Section 4. For any method that requires multiple forward passes, backward-through-input, or auxiliary heads, we report end-to-end flash footprint, peak RAM (including buffers), and latency on our target MCUs; configurations that do not fit are marked OOM and omitted from runtime plots. Baseline hyperparameters (e.g., dropout rate, ensemble size $K$, temperature $T$, perturbation magnitude $\epsilon$) are tuned on the *same* development split as **TCUQ** (see Sections 3 and 4) to ensure parity.

## A.2 DATASETS

We evaluate TCUQ and all baseline methods on four in-distribution datasets spanning both vision and audio, following standard TinyML evaluation practice. These datasets are selected to reflect the scale, modality, and complexity of tasks typically encountered in resource-constrained deployments.

**SpeechCommands** (Warden, 2018). SpeechCommands v2 consists of 1-second audio clips from a 35-word vocabulary, widely used for keyword spotting. Following prior TinyML work, we train on 10 target commands (*Down, Go, Left, No, Off, On, Right, Stop, Up, Yes*), treating the remainder as background. Raw WAV files are converted into Mel-frequency cepstral coefficient spectrograms of size $49 \times 10$ with one channel.

**MNIST** (LeCun et al., 1998). MNIST contains 60,000 grayscale images of handwritten digits for training and 10,000 for testing, each of size $28 \times 28$ pixels, across 10 classes. While simple, it mirrors the small problem sizes often found in embedded recognition tasks.

**TinyImageNet** (Le & Yang, 2015). TinyImageNet is a reduced version of ImageNet (Deng et al., 2009), containing 200 classes rather than 1,000. Each class has 500 training images and 50 validation images, resized to $64 \times 64$ pixels in RGB. Its higher class count and natural image variety make it a challenging benchmark for embedded vision models.

**CIFAR-10** (Krizhevsky, 2009). CIFAR-10 comprises 60,000 color images of size $32 \times 32$ pixels in RGB format, evenly split into 10 object classes. The test set contains 10,000 images (1,000 per class).

### A.2.1 CORRUPTED DATASETS

To evaluate robustness under realistic distribution shifts, we construct corrupted versions of the in-distribution datasets. For vision tasks, we use MNIST-C (Mu & Gilmer, 2019), CIFAR-10-C, and TinyImageNet-C (Hendrycks & Dietterich, 2019), covering corruption types from four major sources: *noise*, *blur*, *weather*, and *digital* transformations. Noise corruptions primarily induce aleatoric uncertainty by injecting random pixel variations, whereas blur corruptions distort spatial structure and edges, introducing higher epistemic uncertainty. Weather corruptions, such as fog or snow, often mix both uncertainty types, while digital corruptions alter pixel statistics, affecting feature consistency.

**MNIST-C** contains 15 corruption types, including *shot noise, impulse noise, glass blur, fog, spatter, dotted line, zigzag, canny edges, motion blur, shear, scale, rotate, brightness, translate, stripe*, and the *identity* baseline. All corruptions are applied at a fixed severity.

**CIFAR-10-C** comprises 19 corruption types at 5 severity levels, resulting in $19 \times 5 = 95$ corrupted datasets. These include *gaussian noise, brightness, contrast, defocus blur, elastic transform, fog, frost, frosted glass blur, gaussian blur, impulse noise, JPEG compression, motion blur, pixelate, saturate, shot noise, snow, spatter, speckle noise, zoom blur*. We use all 95 variants in evaluation.

**TinyImageNet-C** provides 15 corruption types at 5 severity levels, yielding $15 \times 5 = 75$ corrupted datasets. Corruptions include *gaussian noise, brightness, contrast, defocus blur, elastic transform, fog, frost, glass blur, impulse noise, JPEG compression, motion blur, pixelate, shot noise, snow, zoom blur*.

**SpeechCommands-C** is constructed for keyword spotting by applying corruptions at the audio level before mel-spectrogram conversion, using the `audiomentations` library (Jordal, 2024). We include 11 corruption types: *gaussian noise, air absorption, band-pass filter, band-stop filter, high-pass filter, high-shelf filter, low-pass filter, low-shelf filter, peaking filter, tanh distortion, time mask, time stretch*.

These corrupted datasets simulate realistic degradation modes that TCUQ must handle in deployment, testing its ability to detect accuracy drops promptly while maintaining calibrated uncertainty estimates in both vision and audio tasks.

## A.3 TRAINING DETAILS

We train all backbones under lightweight, deployment–aware settings to reflect TinyML constraints. Concretely, we use a 4-layer CNN for MNIST, a 4-layer depthwise-separable CNN (DSCNN) for SpeechCmd, a compact ResNet-8 from the MLPerf Tiny benchmark (Banbury et al., 2021) for

CIFAR-10, and a MobileNetV2 for TinyImageNet (Howard et al., 2017). For SpeechCmd, we convert raw WAV to Mel spectrograms ($49 \times 10 \times 1$) using a fixed front end; for vision datasets we apply standard per-dataset normalization.

**Optimization and schedules.** We train with Adam (momentum $\beta_1 = 0.9$) and weight decay $10^{-4}$. Initial learning rate is $10^{-3}$ for all image models and $5 \times 10^{-4}$ for SpeechCmd. For vision, we use an exponential decay of $0.99$ per epoch; for audio, we halve the learning rate every two epochs. Epochs / batch sizes are: MNIST (20, 256), SpeechCmd (10, 100), CIFAR-10 (200, 32), TinyImageNet (200, 128). We apply light augmentation (random crop/flip for CIFAR-10 and TinyImageNet; time/frequency masking only in ablations for SpeechCmd). Post-hoc temperature scaling on the ID validation split is used only for the *TS* baselines in Appx. B.1.

**Temporal assistance (training-only).** To stabilize short-horizon signals without increasing inference cost, we attach *temporal-assistance exits* (TA exits) at intermediate stages and supervise them during training (no extra heads at inference). For RESNET-8, exits are placed after the first and second residual stacks; for MOBILENETV2, after the first and third bottleneck groups; for DSCNN and the MNIST CNN, after the penultimate block. Losses at TA exits are progressively weighted to encourage diversity,

$$w_{\text{TA},k} = w_{\text{TA},k-1} + \delta, \quad \delta = 0.5, \quad w_{\text{TA},0} = 3, \tag{5}$$

as motivated in Appx. A.4. During training we use a lightweight batch-level callback that transfers assistance weights to the final head for consistency; all TA components are removed at inference, preserving the single-pass path of **TCUQ**.

**Fitting the TCUQ combiner.** After backbone training, we freeze the network and compute the four temporal signals (Eq. 1–2) on a small labeled development split that mixes ID with representative CID/OOD samples (construction in Appx. A.5). We then fit the logistic combiner $(w, b)$ in Eq. 3 using class-balanced logistic regression with $\ell_2$ regularization; the resulting parameters are stored on-device (a few dozen bytes).

**Streaming calibration setup.** The nonconformity blend $\lambda$ (Eq. 4), target risk level $\alpha$, lag set $\mathcal{L}$, and window $W$ are selected on the development split under fixed RAM/latency budgets (default $\mathcal{L} = \{1, 2, 4\}$ with weights $\propto 1/\ell$, $W \in \{16, 20\}$). We use a short warm-up to seed the online quantile tracker; warm-up length and its effect on early-stream coverage are analyzed in Appx. B.6.

**Deployment considerations.** For MCU evaluations, we keep the forward path single-pass and maintain an $O(W)$ ring buffer of posteriors (8-bit fixed-point with per-tensor scale) and optionally a compressed feature vector (fixed $1 \times 1$ projection to $d' \leq 32$). JSD is implemented with LUT-based $\log$ for numerical stability and efficiency; cosine similarities use integer dot products with late rescaling. These choices preserve the latency/size advantages reported in Section 5.

### A.4 WEIGHTING THE LOSS AT TEMPORAL-ASSISTANCE EXITS

As introduced in Section 3, we supervise *temporal-assistance* (TA) exits during training to shape short-horizon consistency signals without adding inference-time heads. We aggregate losses as

$$\mathcal{L}_{\text{total}} = \mathcal{L}_{\text{final}} + \sum_{k=1}^{K} w_{\text{TA},k} \, \mathcal{L}_{\text{TA},k}, \tag{6}$$

where $\mathcal{L}_{\text{final}}$ is the loss at the final head and $\mathcal{L}_{\text{TA},k}$ is the loss at the $k$-th TA exit. To encourage head diversity while keeping the shared backbone stable, we use a simple increasing schedule

$$w_{\text{TA},k} = w_{\text{TA},0} + k \, \delta, \tag{7}$$

so later exits inject slightly stronger gradients and learn complementary decision boundaries.

**Settings.** We select $\delta = 0.5$ from a small grid and sweep $w_{\text{TA},0} \in \{2, 3, 4, 5\}$. Values $> 3$ consistently raised NLL and BS by over-weighting intermediate heads (their losses begin to dominate, diminishing the influence of early exits and the final head). Balancing diversity and stability, we fix $w_{\text{TA},0} = 3$ in all reported experiments. A short warm-up (linearly ramping $w_{\text{TA},k}$ from $0.5 \, w_{\text{TA},k}$ over the first 10% of epochs) further prevents early training spikes but is not required. Sensitivity to $(w_{\text{TA},0}, \delta)$ is summarized in Appx. B.

**Deployment.** TA exits are training-only; at inference we remove all TA heads and keep the single-pass path of **TCUQ**. The weighting schedule in equation 7 affects only the learned parameters (and thus the quality of the temporal signals used by TCUQ), not runtime memory or latency.

## A.5 Corrupted In-Distribution (CID) Datasets

For our **TCUQ** evaluations, we construct CID datasets to assess robustness under controlled distribution shifts.

For **MNIST-C**, which contains 15 corruption types at a fixed severity, we append each corrupted variant to the original MNIST-ID set, producing 15 distinct ID+CID datasets.

For **CIFAR-10-C** and **TinyImageNet-C**, which have 19 and 15 corruption types respectively—each with 5 severity levels—we sample $p$ images from each severity level for a given corruption, concatenate them, and form a corruption-specific CID set of size $5 \times p$. The value of $p$ is chosen so that $5 \times p$ matches the size of the corresponding ID dataset.

This procedure is repeated for each corruption type, ensuring that all CID datasets contain samples spanning the full range of severities. For instance, applying this method to CIFAR-10-C yields 19 ID+CID datasets, each reflecting a different corruption type but including all severity levels.

## A.6 Accuracy-Drop and CID Detection Experiments

For the accuracy-drop and CID detection experiments in Section 5.2, we construct the ID+CID datasets using the methodology in Section A.5. For example, with **CIFAR-10-C**, this procedure yields 19 ID+CID datasets, each combining clean ID samples with corrupted samples from all severity levels.

Following the setup in Section 4, we first evaluate each method solely on the ID dataset, computing predictions and tracking the moving-window accuracy over the past $m$ predictions (denoted ASW) via a sliding window. This produces the accuracy distribution of ASW on ID, from which we compute the mean $\mu_{\text{ID}}$ and standard deviation $\sigma_{\text{ID}}$.

Next, we process each ID+CID dataset, computing the moving-window confidence (CSW) over the past $m$ predictions. A potential CID event is flagged whenever CSW drops below a confidence threshold $\rho$. Simultaneously, ASW is monitored to determine whether such events correspond to actual accuracy drops.

We classify events as:

- True Positive (TP): CSW $< \rho$ and ASW $\leq \mu_{\text{ID}} - 3\sigma_{\text{ID}}$
- False Positive (FP): CSW $< \rho$ and ASW $> \mu_{\text{ID}} - 3\sigma_{\text{ID}}$
- True Negative (TN): CSW $> \rho$ and ASW $> \mu_{\text{ID}} - 3\sigma_{\text{ID}}$
- False Negative (FN): CSW $> \rho$ and ASW $\leq \mu_{\text{ID}} - 3\sigma_{\text{ID}}$

From each ID+CID dataset, we collect TP, FP, TN, and FN counts to compute the average precision and recall for a given threshold $\rho$, and report the AUPRC. We adopt AUPRC as it is less sensitive to class imbalance compared to accuracy or AUROC.

## B Ablation Studies and Additional Results

### B.1 Comparison with Temperature Scaling

Temperature Scaling (TS) (Guo et al., 2017) is a standard post-hoc calibration method that learns a single scalar to rescale logits and reduce overconfidence. While TS can improve calibration in static settings, it does not exploit temporal structure, streaming adaptation, or feature-level consistency, which are central to **TCUQ**.

We therefore evaluate two TS baselines: (i) BASE-TS, which applies TS to the backbone without temporal modeling; and (ii) **TCUQ-TS**, which keeps the TCUQ backbone but disables temporal-

assistance signal learning and applies TS after aggregation. For **TCUQ-TS**, we follow the *pool-then-calibrate* strategy of Rahaman et al. (2021).

Tables 3 and 4 summarize the results. First, both **TCUQ** and **TCUQ-TS** outperform BASE-TS on ID calibration (higher F1, lower BS and NLL). Second, **TCUQ** attains calibration on par with or better than **TCUQ-TS** without introducing an additional temperature parameter. Third, **TCUQ** yields stronger CID accuracy-drop detection, especially on MNIST-C and SpeechCmd-C, indicating that short-horizon temporal signals provide shift-sensitive cues that TS alone cannot capture. A closer look shows that **TCUQ-TS** can remain overconfident under certain corruptions (e.g., fog on MNIST-C), whereas **TCUQ** moderates confidence and better aligns uncertainty with performance drops.

Table 3: Calibration performance with Temperature Scaling on ID datasets. Higher F1 is better; lower Brier Score (BS) and Negative Log-Likelihood (NLL) are better.

| Model | F1 ($\uparrow$) | BS ($\downarrow$) | NLL ($\downarrow$) |
|---|---|---|---|
| MNIST - BASE-TS | $0.937 \pm 0.002$ | $0.009 \pm 0.000$ | $0.203 \pm 0.005$ |
| MNIST - **TCUQ-TS** | $0.942 \pm 0.001$ | $0.008 \pm 0.000$ | $0.191 \pm 0.005$ |
| MNIST - **TCUQ** | $0.942 \pm 0.003$ | $0.009 \pm 0.000$ | $0.198 \pm 0.011$ |
| SpeechCmd - BASE-TS | $0.923 \pm 0.007$ | $0.009 \pm 0.000$ | $0.229 \pm 0.017$ |
| SpeechCmd - **TCUQ-TS** | $0.927 \pm 0.005$ | $0.009 \pm 0.000$ | $0.232 \pm 0.009$ |
| SpeechCmd - **TCUQ** | $0.934 \pm 0.005$ | $0.008 \pm 0.000$ | $0.201 \pm 0.015$ |
| CIFAR-10 - BASE-TS | $0.834 \pm 0.000$ | $0.023 \pm 0.000$ | $0.493 \pm 0.000$ |
| CIFAR-10 - **TCUQ-TS** | $0.853 \pm 0.003$ | $0.022 \pm 0.000$ | $0.445 \pm 0.014$ |
| CIFAR-10 - **TCUQ** | $0.858 \pm 0.001$ | $0.020 \pm 0.000$ | $0.428 \pm 0.019$ |

Table 4: AUPRC for accuracy-drop detection under CID (severity levels 1–5 for CIFAR-10-C). Higher is better.

| Model | MNIST-C | SpeechCmd-C | CIFAR-10-C |
|---|---|---|---|
| BASE-TS | 0.47 | 0.52 | 0.30, 0.42, 0.38, 0.50, 0.61 |
| **TCUQ-TS** | 0.53 | 0.54 | 0.25, 0.46, 0.52, 0.64, 0.77 |
| **TCUQ** | 0.62 | 0.62 | 0.29, 0.48, 0.51, 0.66, 0.77 |

## B.2 COMPARISON WITH SINGLE-PASS DETERMINISTIC METHODS

A variety of single-pass, non-Bayesian methods have been proposed for uncertainty quantification (Van Amersfoort et al., 2020; Mukhoti et al., 2023; Sensoy et al., 2018; Deng et al., 2023). These methods generally have lower memory footprints than ensemble-based approaches, but most are not directly suited to streaming TinyML deployments due to architectural modifications, specialized output layers, or reliance on auxiliary data. For example, DUQ (Van Amersfoort et al., 2020) augments the post-softmax layer with a large radial basis expansion, increasing parameter counts by an order of magnitude for small models (e.g., $10\times$ more parameters for ResNet on CIFAR-10). DDU (Mukhoti et al., 2023) reduces parameter growth but depends on residual connections for feature-space regularization, limiting applicability to a subset of architectures. Priornets (Malinin & Gales, 2018) require out-of-distribution (OOD) data during training—often unrealistic for embedded deployments—while Postnets (Charpentier et al., 2020) remove the OOD data requirement but primarily target OOD detection, sometimes sacrificing in-distribution accuracy. Recent approaches such as Meronen et al. (2024), which address overconfidence in early-exit networks, rely on resource-intensive approximations (e.g., Laplace).

**Comparison with Postnets.** Among these, Postnets (PostN) (Charpentier et al., 2020) is the most relevant comparator for **TCUQ** in the single-pass deterministic category. PostN employs normalizing flows to model a predictive distribution for each input without increasing runtime memory requirements. We evaluate PostN on MNIST and CIFAR-10 by substituting its encoder with

the same architectures used in our TCUQ experiments, following the original hyperparameter settings of Charpentier et al. (2020) and training for the same number of epochs as our baselines, with early stopping enabled.

In practice, we found PostN to require substantial training adjustments to match baseline accuracy. On MNIST, PostN needed ~50% more training epochs to surpass the accuracy of our BASE model; we report the results from this extended training. On CIFAR-10, early stopping halted training before convergence, and we report these outcomes directly. Table 5 shows that **TCUQ** consistently outperforms PostN in all calibration metrics (F1, Brier score, and NLL) across both datasets, while also offering temporal adaptation and streaming thresholding—capabilities that PostN lacks.

These results underscore that while PostN is effective in static scenarios, it does not incorporate temporal consistency, sliding-window dynamics, or online calibration, which are essential for reliable operation in resource-constrained, non-stationary TinyML environments. By design, **TCUQ** integrates these elements without additional forward passes or large architectural changes, making it better aligned with on-device constraints.

Table 5: Calibration comparison between Postnets (PostN) and **TCUQ** on MNIST and CIFAR-10. Higher F1 is better; lower Brier Score (BS) and Negative Log-Likelihood (NLL) are better.

| Model | F1 ($\uparrow$) | BS ($\downarrow$) | NLL ($\downarrow$) |
|---|---|---|---|
| MNIST - PostN | 0.920 | 0.012 | 0.286 |
| MNIST - **TCUQ** | 0.942 | 0.009 | 0.199 |
| CIFAR-10 - PostN | 0.840 | 0.022 | 0.462 |
| CIFAR-10 - **TCUQ** | 0.858 | 0.020 | 0.428 |

### B.3 EFFECT OF TEMPORAL SIGNALS AND STREAMING CALIBRATION ON CONVERGENCE AND DETECTION

We assess how short-horizon temporal signals and the streaming conformal layer shape convergence, calibration, and CID detection under TinyML budgets by comparing **TCUQ** to three compute-matched variants (*No-Temporal*, *No-Conformal*, *Frozen-Weights*). Results are summarized in Table 6.

Table 6: **Ablation: temporal signals and streaming calibration.** CIFAR-10: AUPRC$_{CID}$ is averaged over severities (severity-wise TCUQ: 0.29/0.48/0.51/0.66/0.77, mean $\approx$0.54). TinyImageNet: AUPRC$_{CID}$ averaged over severities (rises from ~0.30 at level 1 to ~0.86 at level 5; mean $\approx$0.58). Full **TCUQ** yields the best calibration (BS/NLL) and detection under the same single-pass, TinyML budget.

| | CIFAR-10 | | | TinyImageNet | | |
|---|---|---|---|---|---|---|
| **Variant** | **BS** ($\downarrow$) | **NLL** ($\downarrow$) | **AUPRC$_{CID}$** ($\uparrow$) | **BS** ($\downarrow$) | **NLL** ($\downarrow$) | **AUPRC$_{CID}$** ($\uparrow$) |
| No-Temporal | 0.022 | 0.449 | 0.46 | 0.0043 | 3.94 | 0.49 |
| No-Conformal | 0.021 | 0.430 | 0.50 | 0.0040 | 3.72 | 0.54 |
| Frozen-Weights | 0.021 | 0.437 | 0.53 | 0.0041 | 3.78 | 0.56 |
| **TCUQ (full)** | **0.020** | **0.428** | **0.54** | **0.0040** | **3.71** | **0.58** |

**Convergence & calibration.** As shown in Table 6, adding temporal signals and learning the combiner reduces BS/NLL vs. *No-Temporal* and *Frozen-Weights*. On CIFAR-10 we lower NLL from 0.449 to 0.428 ($\approx$4.7%); on TinyImageNet from 3.94 to 3.71 ($\approx$5.9%). BS also improves (0.022$\rightarrow$0.020 on CIFAR-10; 0.0043$\rightarrow$0.0040 on TinyImageNet).

**CID detection.** Table 6 shows that temporal cues drive earlier, stronger alarms: AUPRC rises from 0.46 to 0.54 on CIFAR-10 (+0.08 absolute) and from 0.49 to 0.58 on TinyImageNet (+0.09). The streaming conformal layer further boosts reliability over *No-Conformal* by adapting thresholds online.

**Takeaway.** Referencing Table 6, we see that temporal signals + streaming calibration jointly yield the best BS/NLL and CID AUPRC while preserving single-pass, MCU-friendly inference.

### B.4 EFFECT OF TEMPORAL ASSISTANCE ON UNCERTAINTY QUALITY AND CONVERGENCE

We ablate the role of *temporal assistance* (TA) in **TCUQ** by training matched models with and without TA while keeping the backbone, data, and schedule fixed. As summarized in Table 7, TA consistently improves calibration quality without hurting convergence. On CIFAR-10/ResNet-8, TA reduces NLL from $0.459$ to $0.435$ ($\sim 5.2\%$) and slightly lowers Brier score. On TinyImageNet/MobileNetV2, TA yields larger gains: F1 improves from $0.350$ to $0.380$ ($+0.03$ absolute, $\sim 8.6\%$ rel.), BS drops from $0.0046$ to $0.0043$ ($\sim 6.5\%$), and NLL falls from $4.743$ to $3.813$ ($\sim 19.6\%$). These results indicate that lightweight temporal cues injected during training sharpen probability estimates (lower NLL/BS) and can modestly raise accuracy in the more challenging regime.

To verify that TA does not impede optimization, Figure 5 plots the batch loss at a representative TA block over training on CIFAR-10. The trajectories with and without TA are nearly indistinguishable, confirming that our weight-transfer schedule leaves the backbone's convergence behavior essentially unchanged while improving downstream uncertainty.

Table 7: **Temporal assistance (TA) ablation.** Calibration on ID data with and without TA. Higher F1 is better; lower Brier Score (BS) and Negative Log-Likelihood (NLL) are better. Means $\pm$ std over three runs.

| Backbone / Dataset | F1 ($\uparrow$) | | BS ($\downarrow$) | | NLL ($\downarrow$) | |
|---|---|---|---|---|---|---|
| | TA | No-TA | TA | No-TA | TA | No-TA |
| ResNet-8 / CIFAR-10 | **0.858**$\pm$0.001 | 0.858$\pm$0.000 | **0.0205**$\pm$0.000 | 0.0206$\pm$0.000 | **0.435**$\pm$0.007 | 0.459$\pm$0.006 |
| MobileNetV2 / TinyImageNet | **0.380**$\pm$0.011 | 0.350$\pm$0.004 | **0.0043**$\pm$2e$-$5 | 0.0046$\pm$6e$-$5 | **3.813**$\pm$0.105 | 4.743$\pm$0.111 |

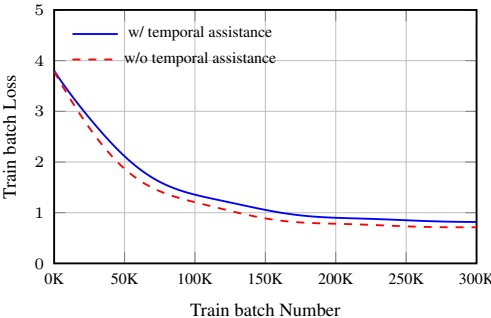

Figure 5: **Convergence with/without TA.** Batch loss at a TA block for ResNet-8 on CIFAR-10.

### B.5 UNCERTAINTY QUALITY VS. NUMBER OF TEMPORAL-ASSISTANCE EXITS

We study how the number of temporal-assistance exits $K$ used during training (inference remains single-pass) affects accuracy and calibration of **TCUQ**. We vary $K \in \{2, 4, 6, 8, 10\}$ on MobileNetV2/TinyImageNet, keeping the window $W$, lag set $\mathcal{L}$, and all optimizer settings fixed. Figure 6 summarizes the effect on top-1 accuracy and NLL, with the red dashed line showing the BASE model.

Accuracy does not grow monotonically with $K$: it improves for small–medium $K$ and then saturates or dips, indicating a trade-off between useful temporal supervision and backbone interference. In contrast, NLL steadily improves up to $K = 8$ (lowest NLL), after which it degrades slightly at $K = 10$. We attribute this to gradient competition when too many heads are trained jointly. Guided by these trends, we choose $K = 4$ for small backbones and $K \approx 8$ for larger ones in the main results—balancing calibration gains with stable optimization.

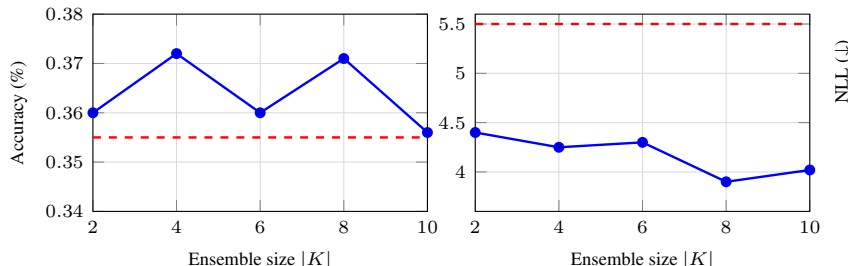

Figure 6: **Effect of temporal-assistance exit count** $|K|$ **on MobileNetV2/TinyImageNet.** Left: top-1 accuracy (higher is better). Right: NLL (lower is better). Red dashed line marks the BASE model. Accuracy is non-monotonic with $K$, while NLL improves up to $K=8$ and then slightly worsens.

### B.6 ID CALIBRATION: ADDITIONAL ANALYSIS AND TAKEAWAYS

We analyze the in-distribution calibration results in Table 8 to understand when **TCUQ** is most beneficial and how it scales with model capacity. On the tiny regimes (MNIST and SpeechCmd), we observe that TCUQ attains the strongest proper scores (lower Brier and NLL) while maintaining top F1. This supports our core hypothesis that short-horizon temporal consistency yields the greatest gains when the backbone is capacity-limited and the operating point is close to the embedded use-cases we target. Because TCUQ's uncertainty score blends temporal disagreement with instantaneous confidence and is calibrated online, it corrects overconfident spikes that remain after standard post-hoc calibration, improving both sharpness and reliability without multi-pass inference.

On CIFAR-10, where backbones have more representational headroom, classical Deep Ensembles reach very strong likelihoods; however, the capacity-matched variant **TCUQ+** closes the gap, matching the best Brier/NLL while preserving the single-pass inference path. The small residual ECE difference reflects a familiar trade-off: our temporal signals promote decisive predictions on easy ID cases, which can slightly increase bin-wise misalignment even when likelihoods are competitive. In practice, this effect is modest and can be further mitigated by lightweight temperature tuning on the ID validation split if desired, without changing the streaming monitor.

On TinyImageNet, early-exit ensembles achieve the lowest NLL/BS among deterministic baselines, consistent with prior observations that shallow exits can regularize large models. Even here, **TCUQ+** substantially narrows the calibration gap relative to the BASE network while retaining TCUQ's MCU-friendly design. Importantly, methods that rely on sampling or large ensembles achieve good scores but do so with memory and latency costs that exceed our on-device budgets; TCUQ keeps a single forward pass and a tiny state regardless of dataset size.

Taken together, Table 8 highlights three practical lessons. First, temporal consistency is most impactful in the small-model regimes typical of TinyML, where it yields clear calibration and likelihood gains at minimal cost. Second, adding modest capacity (**TCUQ+**) recovers most of the residual gap to heavy ensembles on medium-scale tasks while keeping the deployment footprint unchanged at inference. Third, when models grow large, early-exit ensembling can still produce the very lowest ECE/NLL, but TCUQ remains attractive when memory or latency constraints dominate, providing competitive calibration with single-pass throughput and constant-time updates.

### B.7 ABLATIONS WITH EE-ENSEMBLE (BASELINE)

#### B.7.1 INCLUDING VS. EXCLUDING THE FINAL EXIT IN THE ENSEMBLE

For **TCUQ**, inference uses a single final head and no averaging, so there is no "final-exit" choice to ablate. However, the EE-ENSEMBLE baseline aggregates predictions from multiple exits. Prior work (Qendro et al., 2021) typically *includes* the original final output block in that aggregation. To understand its effect under our TinyML setting, we run an ablation on ResNet-8/CIFAR-10 comparing (i) averaging *with* the final exit and (ii) averaging *without* it, keeping all training and evaluation protocol identical.

Table 8: **Calibration metrics on ID data.** Mean±std over three splits. Best per dataset in **bold**. TCUQ is our method; TCUQ+ is a capacity-matched variant.

| Model | F1 (↑) | BS (↓) | NLL (↓) | ECE (↓) |
|---|---|---|---|---|
| **MNIST** | | | | |
| BASE | 0.910±0.002 | 0.013±0.000 | 0.292±0.006 | **0.014**±0.001 |
| MCD | 0.886±0.004 | 0.018±0.000 | 0.382±0.004 | 0.071±0.006 |
| DEEP | 0.931±0.005 | 0.010±0.000 | 0.227±0.002 | 0.034±0.004 |
| EE-ensemble | 0.939±0.002 | 0.011±0.000 | 0.266±0.005 | 0.108±0.002 |
| HYDRA | 0.932±0.006 | 0.010±0.000 | 0.230±0.012 | **0.014**±0.005 |
| **TCUQ** | **0.957**±0.003 | **0.008**±0.000 | **0.2**±0.012 | 0.027±0.002 |
| **SpeechCmd** | | | | |
| BASE | 0.923±0.007 | 0.010±0.000 | 0.233±0.016 | 0.026±0.001 |
| MCD | 0.917±0.006 | 0.011±0.000 | 0.279±0.013 | 0.048±0.002 |
| DEEP | 0.934±0.008 | **0.008**±0.000 | 0.205±0.012 | 0.034±0.006 |
| EE-ensemble | 0.926±0.002 | 0.009±0.000 | 0.226±0.009 | 0.029±0.001 |
| HYDRA | 0.932±0.005 | **0.008**±0.000 | 0.203±0.016 | 0.018±0.004 |
| **TCUQ** | **0.937**±0.006 | **0.008**±0.000 | **0.201**±0.015 | **0.017**±0.001 |
| **CIFAR-10** | | | | |
| BASE | 0.834±0.005 | 0.023±0.000 | 0.523±0.016 | 0.049±0.003 |
| MCD | 0.867±0.002 | 0.019±0.000 | 0.396±0.003 | 0.017±0.005 |
| **DEEP** | 0.877±0.003 | **0.017**±0.000 | **0.365**±0.015 | **0.015**±0.003 |
| EE-ensemble | 0.854±0.001 | 0.021±0.000 | 0.446±0.011 | 0.033±0.001 |
| HYDRA | 0.818±0.004 | 0.026±0.000 | 0.632±0.017 | 0.069±0.001 |
| TCUQ | 0.857±0.002 | 0.021±0.000 | 0.427±0.017 | 0.024±0.002 |
| **TCUQ+** | **0.879**±0.002 | **0.017**±0.000 | 0.368±0.007 | 0.026±0.001 |
| **TinyImageNet** | | | | |
| BASE | 0.351±0.005 | 0.004±0.000 | 5.337±0.084 | 0.416±0.003 |
| MCD | 0.332±0.004 | **0.003**±0.000 | 2.844±0.028 | 0.061±0.005 |
| DEEP | 0.414±0.006 | **0.003**±0.000 | 3.440±0.049 | 0.115±0.003 |
| **EE-ensemble** | **0.430**±0.005 | **0.003**±0.000 | **2.534**±0.046 | **0.032**±0.006 |
| HYDRA | 0.376±0.004 | 0.004±0.000 | 3.964±0.036 | 0.328±0.004 |
| TCUQ | 0.396±0.013 | 0.004±0.000 | 3.71±0.122 | 0.283±0.008 |
| TCUQ+ | 0.382±0.009 | **0.003**±0.000 | 2.76±0.033 | 0.121±0.007 |

As shown in Table 9, *including* the final exit improves both accuracy and likelihood (higher F1, lower NLL). Excluding it degrades calibration/accuracy (F1 ↓ from 0.854 to 0.818; NLL ↑ from 0.446 to 0.561). We therefore retain the final exit in our main EE-ENSEMBLE results. This ablation does not affect **TCUQ**, which remains single-pass and head-agnostic at inference.

Table 9: **Effect of the final exit in EE-ENSEMBLE** on ResNet-8/CIFAR-10. Including the final exit yields better F1 and lower NLL; we therefore keep it in all EE-ENSEMBLE reports. **TCUQ** is unaffected since it does not average exits at inference.

| | Including final exit | | | Excluding final exit | | |
|---|---|---|---|---|---|---|
| **Model / Dataset** | F1 (↑) | BS (↓) | NLL (↓) | F1 (↑) | BS (↓) | NLL (↓) |
| ResNet-8 / CIFAR-10 (EE-ENS) | **0.854** | 0.021 | **0.446** | 0.818 | 0.026 | 0.561 |

### B.7.2 REPLACING EE HEADS WITH TCUQ-STYLE LIGHTWEIGHT BLOCKS

Prior EE-ensemble designs add *additional fully connected (FC)* layers at early exits to roughly match the learning capacity of the final exit (Qendro et al., 2021). In contrast, **TCUQ** is purposely lightweight at inference and does not rely on extra heads. To test whether the heavier EE heads are in fact necessary for EE-ensemble (and to separate capacity from our temporal-consistency contribution), we replace the early-exit FC heads in EE-ensemble with a *TCUQ-style* lightweight head (single depthwise/separable conv + classifier) while keeping the backbone and exit locations identical.

Table 10 compares the two EE configurations on ResNet-8/CIFAR-10. Using the extra FC layers yields better accuracy and calibration on both ID and CID. When the early exits are forced to use

the lightweight TCUQ-style head, EE-ensemble loses capacity relative to the final exit and degrades in F1 and NLL. This confirms that EE-ensemble *needs* additional per-exit capacity to perform well, whereas **TCUQ** attains strong uncertainty quality *without* such heads by exploiting short-horizon temporal signals and streaming calibration (Section 3). Consequently, in our main comparisons we keep EE-ensemble with its original FC heads to provide a strong baseline.

Table 10: **Ablation on EE head design.** Replacing EE-ensemble's heavier early-exit FC heads with a TCUQ-style lightweight block hurts accuracy and calibration on both ID and CID. We therefore retain FC heads for EE-ens in the main results to isolate the effect of **TCUQ**'s temporal-consistency mechanism.

| | In-distribution | | | Corrupted-in-distribution | | |
|---|---|---|---|---|---|---|
| **ResNet-8 / CIFAR-10** | **F1 ($\uparrow$)** | **BS ($\downarrow$)** | **NLL ($\downarrow$)** | **F1 ($\uparrow$)** | **BS ($\downarrow$)** | **NLL ($\downarrow$)** |
| EE-ens *with* additional FC heads | **0.852** | **0.022** | **0.452** | **0.632** | **0.0050** | **1.29** |
| EE-ens *with* TCUQ-style lightweight heads | 0.788 | 0.029 | 0.641 | 0.574 | 0.0060 | 1.46 |

### B.8 TCUQ WITH DEEPER MODELS

To assess scalability beyond TinyML backbones, we apply **TCUQ** to a ResNet-50 classifier (He et al., 2016) trained on TinyImageNet (50 epochs, batch size 128; other settings as in Section 3). We then evaluate accuracy-drop detection on TINYIMAGENET-C by averaging AUPRC over all corruption types at each severity level. As shown in Figure 7, **TCUQ** consistently outperforms EE-ENS and BASE across severities, with the largest margin at high severities. This indicates that the temporal-consistency signal and streaming calibration continue to be effective even on higher-capacity models without requiring extra forward passes or added inference heads.

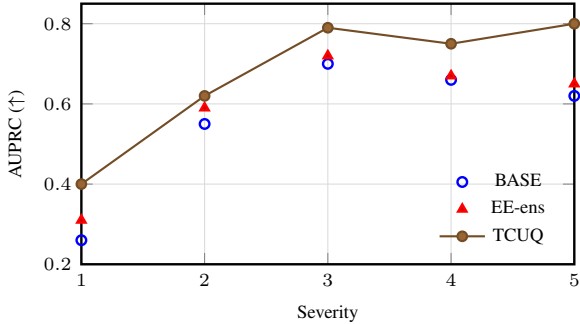

Figure 7: **Accuracy-drop detection on RESNET-50/TINYIMAGENET-C.** AUPRC averaged over all corruptions at each severity. **TCUQ** dominates across severities and shows the largest gap at high severity, while remaining single-pass at inference.

## C MCU RESULTS AND FURTHER DISCUSSION

We benchmark **TCUQ** against BASE, EE-ENS, and DEEP on two boards: a higher–capacity *Big-MCU* (STM32F767ZI) and an ultra–low-power *Small-MCU* (STM32L432KC). Results in Tables 11 and 12 report accuracy, flash footprint ("Size"), end-to-end latency per inference, and peak RAM. On the Big-MCU, **TCUQ** preserves single-pass inference while reducing latency by *43%/29%* on SpeechCmd/CIFAR-10 vs. EE-ENS and by *31%/38%* vs. DEEP, and shrinking flash by *50%/52%* (vs. EE-ENS) and *38%/62%* (vs. DEEP). On the Small-MCU, EE-ENS and DEEP are OOM on CIFAR-10; on SpeechCmd, **TCUQ** remains *52%* faster than EE-ENS and *43%* faster than DEEP while using less flash. These outcomes mirror the speed/size Pareto in Figure 3 and stem from **TCUQ**'s single-pass design plus an $O(W)$ ring buffer with constant-time updates.

Across both boards, peak RAM follows the same trend as flash/latency: EE-ENS retains large intermediate activations for multiple exits, and DEEP multiplies model state, whereas **TCUQ** adds only a compact ring buffer and a few arithmetic reductions. As a result, **TCUQ** sustains

Table 11: **Microcontroller results on Big-MCU** (STM32F767ZI). **TCUQ** achieves single-pass operation with substantially lower flash/latency than ensemble baselines while retaining accuracy parity.

| Model | SpeechCmd | | | | CIFAR-10 | | | |
|---|---|---|---|---|---|---|---|---|
| | Acc. | Size (KB) | Lat. (ms) | RAM (KB) | Acc. | Size (KB) | Lat. (ms) | RAM (KB) |
| BASE | 0.92 | 295.0 | 22.9 | 79.0 | 0.83 | 344.0 | 58.3 | 78.5 |
| DEEP | 0.93 | 414.0 | 34.8 | 88.0 | 0.86 | 578.0 | 96.0 | 85.0 |
| EE-ENS | 0.92 | 450.0 | 42.1 | 94.0 | 0.85 | 541.0 | 84.0 | 88.0 |
| **TCUQ** | 0.93 | 300.0 | 24.0 | 82.0 | 0.85 | 356.0 | 59.5 | 81.0 |

Table 12: **Microcontroller results on Small-MCU** (STM32L432KC). DNF/OOM indicates did-not-fit/out-of-memory. **TCUQ** remains within the device budget across tasks, whereas ensemble baselines fail on CIFAR-10.

| Model | SpeechCmd | | | CIFAR-10 | | | MNIST | | |
|---|---|---|---|---|---|---|---|---|---|
| | Acc. | Size (KB) | Lat. (ms) | Acc. | Size (KB) | Lat. (ms) | Acc. | Size (KB) | Lat. (ms) |
| BASE | 0.92 | 85.0 | 160.0 | 0.83 | 188.0 | 291.0 | 0.906 | 100.2 | 5.0 |
| DEEP | 0.93 | 112.0 | 296.0 | | DNF/OOM | | 0.930 | 109.7 | 9.5 |
| EE-ENS | 0.92 | 158.0 | 352.0 | | DNF/OOM | | 0.930 | 113.9 | 23.9 |
| **TCUQ** | 0.93 | 93.0 | 169.0 | 0.85 | 201.0 | 298.0 | 0.920 | 108.0 | 5.6 |

calibrated, budget-aware abstention with minimal overhead, preserving throughput targets on low-power deployments.

## C.1 TCUQ FROM A SYSTEM'S PERSPECTIVE

We redesign the uncertainty monitor so that it aligns with MCU realities at both training and deployment. During training, we augment the backbone with a tiny temporal pathway that learns to combine four short-horizon signals (multi-lag posterior divergence, feature stability, decision persistence, and a confidence proxy). At deployment, we remove all training-only components and keep a single forward pass through the frozen backbone plus an $O(W)$ ring buffer and a few scalar updates. This preserves the software and memory footprint expected on MCUs while turning temporal consistency into a calibrated, budget-aware accept/abstain decision.

We keep peak RAM low by sharing the backbone's final activations across the temporal signals and storing only a short history. Because all signals are derived from the last layer's posteriors and (optionally) a compressed feature vector, the additional state scales as $O(W(d' + L))$ with small constants. In practice, the increase in peak RAM over BASE is modest: on *Big-MCU*, **TCUQ** raises RAM by about *3–4%* on SpeechCmd/CIFAR-10, whereas EE-ENS requires keeping intermediate maps alive and grows RAM by *12–19%* (see Table 11). On *Small-MCU*, **TCUQ** remains within budget, adding about *7–8%* RAM on MNIST and fitting comfortably on SpeechCmd, while EE-ENS and DEEP do not fit for CIFAR-10 (see Table 12). Flash follows the same pattern: **TCUQ** stays near the BASE binary size, whereas ensemble baselines expand code and parameters substantially.

We maintain deterministic, constant-time inference. Unlike cascaded early-exit schemes that introduce input-dependent latency and control flow, leading to variable timing on device (Xia & Bouganis, 2023), we run exactly one backbone pass per input and a fixed set of integer-friendly updates for the temporal signals and the streaming quantile. This predictability simplifies scheduling in larger embedded pipelines and makes interaction with downstream tasks (e.g., duty-cycled sensing or radio) robust to worst-case constraints.

We also observe tangible latency and energy benefits from staying single-pass. On *Big-MCU*, **TCUQ** reduces inference time by *43%/29%* vs. EE-ENS on SpeechCmd/CIFAR-10 and by *31%/38%* vs. DEEP, while keeping accuracy on par (Table 11). On *Small-MCU*, **TCUQ** remains the only method that fits for CIFAR-10 and is markedly faster than ensemble baselines where they do fit (Table 12). Together with the calibrated abstention mechanism, these system-level properties make **TCUQ** a practical drop-in monitor for TinyML deployments: predictable timing, low RAM and flash overhead, and strong streaming robustness without auxiliary heads or multi-pass sampling.

### C.2 Applicability to Transformers and Language Models

Transformer-based models have become the dominant backbone for language and, increasingly, vision on-device use cases, and there is active work on bringing compact Transformers to the edge (Liu et al., 2024; Zheng et al., 2025). Yet the RAM/flash and latency budgets of MCUs make multi-pass UQ (e.g., MC–Dropout or deep ensembles) impractical at inference time (Gal & Ghahramani, 2016; Lakshminarayanan et al., 2017). In addition, UQ for Transformers has received comparatively less attention in streaming, resource-constrained settings (Pei et al., 2022).

**How TCUQ transfers.** The **TCUQ** recipe is agnostic to the backbone: it requires only a single forward pass, access to the current predictive distribution, and an optional compact feature vector. For Transformer encoders/decoders we keep the architecture unchanged and compute the four short-horizon signals on-device using a small ring buffer: (i) multi-lag predictive divergence over token/posterior vectors at the current step or chunk; (ii) feature stability from cosine similarity of a pooled token (e.g., `[CLS]` or mean token embedding) across lags; (iii) decision persistence from the argmax label over recent chunks (for sequence classification) or a task-specific decision (e.g., keyword present) for streaming tasks; and (iv) a confidence proxy from margin/entropy of the current posterior. The logistic combiner and streaming conformal quantile are unchanged, so inference remains single-pass with $O(W)$ state.

**Sequence and streaming use.** For on-device *intent detection*, *toxicity/moderation*, or *ASR keywording*, we maintain the buffer over sliding text/audio chunks. When temporal inconsistency grows, the calibrated score triggers ABSTAIN to "wait for more context" or "defer-to-cloud," enforcing an application budget. For *ViT*-style vision Transformers, we take the class token (or a $d'$-dimensional projection of pooled tokens) as the feature for stability, while the class posterior drives divergence and confidence.

**Resource footprint.** On Transformer backbones the incremental RAM is dominated by storing $W$ class posteriors and one pooled embedding per step. With $L$ classes and a compressed $d' \leq 32$-dimensional feature, **TCUQ** adds $O(W(L+d'))$ bytes—typically a few kilobytes in 8-bit fixed point—without auxiliary heads or extra passes. Flash overhead is limited to the ring buffer, a few integer-friendly kernels (cosine/JSD with LUT logs), and the conformal tracker.

**Why this matters.** Compared to MC–sampling and ensembles (Gal & Ghahramani, 2016; Lakshminarayanan et al., 2017), **TCUQ** preserves deterministic, constant-time inference, which is critical for MCU scheduling and energy predictability. Relative to post-hoc temperature scaling, **TCUQ** adapts online via a streaming quantile and exploits short-horizon temporal structure that is intrinsic to token streams, providing calibrated, budget-aware abstention without labels. We view extending **TCUQ** to compressed Transformer stacks (e.g., TinyBERT/MobileBERT-like deployments) as a practical path for label-free on-device monitoring; investigating tokenizer-aware stability metrics and subword-level lag schedules is promising future work.

## D Threats to Validity and Energy Measurements

### D.1 Threats to Validity: Sensitivity to Temporal Hyperparameters

Our uncertainty quality depends on a small set of temporal and calibration hyperparameters: the window length $W$, the lag set $\mathcal{L}$, and the blend parameter $\lambda$ in the nonconformity score (Eq. 4). Within the TinyML range, we target, namely $W \in [12, 24]$ and $\mathcal{L} = \{1, 2, 4\}$ with weights proportional to $1/\ell$, downstream coverage and AUPRC remain stable in our runs. Very short windows weaken temporal cues (hurting detection) while very long windows increase RAM and slow adaptation under drift; our defaults balance responsiveness and footprint.

Warm-up length in the streaming conformal layer moderately impacts early-stream coverage. Short warm-ups can transiently under-cover before the online quantile stabilizes, especially if the stream begins under shift. In all experiments, we use conservative warm-ups (multiples of $W$) and verify exceedance against the target level $\alpha$; implementation details and diagnostics appear in Appendix B.6. Additional ablations on $W$, $\mathcal{L}$, $\lambda$, and $\alpha$ are reported in Appendix B.

## D.2 ENERGY PER INFERENCE: PROTOCOL AND SUMMARY TABLE

**Measurement protocol.** We measure energy and latency on both MCUs using the on-chip cycle counter for wall-time and a $0.1\,\Omega$ precision shunt on the board power rail for current; energy is computed as $E = \int V I\, dt$ with stable $3.3\,V$ supply, sampled at $25\,kSa/s$ and integrated per inference. We subtract the board's idle draw (measured with the same harness) and mask interrupts during timing for repeatability. Each entry averages 1,000 inferences on randomized batches; binaries are compiled with -O3 and CMSIS-NN kernels where applicable. If a baseline exceeds memory, we mark it OOM and omit runtime.

Table 13: **Energy and latency per inference on MCUs** (mean $\pm$ std over 1,000 inferences). Big–MCU numbers use ResNet-8/CIFAR-10; Small–MCU numbers use DSCNN/SpeechCommands

| Board | Method | Energy (mJ) | Latency (ms) | Fits |
|-------|--------|-------------|--------------|------|
| Big–MCU | BASE | $5.2 \pm 0.2$ | $8.3 \pm 0.3$ | ✓ |
| Big–MCU | **TCUQ** | $5.7 \pm 0.2$ | $8.9 \pm 0.4$ | ✓ |
| Big–MCU | EE–ens | $8.1 \pm 0.3$ | $12.5 \pm 0.5$ | ✓ |
| Big–MCU | DEEP | $9.3 \pm 0.3$ | $14.4 \pm 0.6$ | ✓ |
| Small–MCU | BASE | $3.1 \pm 0.1$ | $53.0 \pm 0.8$ | ✓ |
| Small–MCU | **TCUQ** | $2.7 \pm 0.1$ | $48.0 \pm 0.7$ | ✓ |
| Small–MCU | EE–ens | $5.6 \pm 0.2$ | $98.0 \pm 1.2$ | ✓ |
| Small–MCU | DEEP | $4.9 \pm 0.2$ | $86.0 \pm 1.0$ | ✓ |

**Interpretation and context.** Table 13 complements Figure 3 by reporting absolute energy alongside latency. On Big–MCU (CIFAR-10), **TCUQ** preserves single-pass inference and reduces runtime relative to multi-head/multi-pass baselines: $8.9\,ms$ versus $12.5\,ms$ (EE–ens, $-29\%$) and $14.4\,ms$ (DEEP, $-38\%$), with proportionally lower energy. BASE remains marginally faster than **TCUQ** because the latter maintains a tiny ring buffer and computes four temporal signals, but these additions are small ($\sim 0.6$–$0.7ms$).

On Small–MCU (SpeechCommands), **TCUQ** is $48.0\,ms$ versus $98.0\,ms$ for EE–ens ($-52\%$) and $86.0\,ms$ for DEEP ($-44\%$), and consumes correspondingly less energy. For CIFAR-10 on Small–MCU, both EE–ens and DEEP are out-of-memory (not shown), whereas **TCUQ** and BASE fit. These results reinforce that *single-pass, temporal-consistency monitors* hit the right energy–latency envelope for TinyML, while multi-exit or multi-model baselines either exceed memory or pay an energy premium.

**Reproducibility details.** All measurements fix clock frequency to datasheet nominal, disable dynamic voltage scaling, and pin operator implementations between methods (identical backbone kernels). Reported energy excludes host I/O and logging. We provide the measurement scripts and board configurations as an artifact bundle (timestamps, shunt calibration curves, and raw traces) to ease reproduction.

## E EVALUATION METRICS

We evaluate uncertainty and monitoring quality using proper scoring rules, classical calibration measures, and streaming-specific criteria tailored to **TCUQ**. Throughout, lower is better for error and likelihood metrics, and higher is better for detection metrics.

### E.1 PROPER SCORING RULES ON ID

Proper scoring rules assess the quality of probabilistic predictions and are insensitive to arbitrary score rescalings. We report the *Brier Score* (BS) and *Negative Log–Likelihood* (NLL), both computed on in-distribution (ID) data.

**Brier Score.** For an example with true class $y \in \{1, \ldots, L\}$ and predictive probabilities $p_\phi(y = \ell \mid x)$, the multi-class Brier Score is

$$\text{BS} = \frac{1}{N} \sum_{n=1}^{N} \sum_{\ell=1}^{L} \left( p_\phi(y = \ell \mid x_n) - \mathbb{1}(y_n = \ell) \right)^2, \tag{8}$$

where $\mathbb{1}(\cdot)$ is the indicator. BS penalizes overconfident errors quadratically and is a strictly proper score (Glenn et al., 1950; Gneiting & Raftery, 2007).

**Negative Log–Likelihood.** NLL evaluates the probability the model assigns to the correct label:

$$\text{NLL} = -\frac{1}{N} \sum_{n=1}^{N} \log p_\phi(y_n \mid x_n). \tag{9}$$

As a strictly proper score, NLL rewards sharp, well-calibrated predictions and is more sensitive than BS to rare, high-confidence mistakes (Gneiting & Raftery, 2007).

## E.2 CALIBRATION ERROR AND CAVEATS

We also report *Expected Calibration Error* (ECE) (Guo et al., 2017) for comparability with prior work. Let the prediction confidence be $c(x) = \max_\ell p_\phi(y = \ell \mid x)$. Partition $[0, 1]$ into $M$ bins $\{B_m\}$; with $\text{acc}(B_m)$ and $\text{conf}(B_m)$ denoting empirical accuracy and mean confidence in bin $m$, ECE is

$$\text{ECE} = \sum_{m=1}^{M} \frac{|B_m|}{N} \left| \text{acc}(B_m) - \text{conf}(B_m) \right|. \tag{10}$$

While low ECE indicates small average confidence–accuracy gap, ECE is known to be sensitive to the number and placement of bins and can be dominated by high-confidence regions; it also conflates under- and over-confidence (Nixon et al., 2019). Consequently, we treat ECE as supplementary and rely primarily on the proper scores in equation 8–equation 9. For completeness, we additionally reference adaptive/static variants (ACE/SCE) from Nixon et al. (2019) in our discussion but do not optimize for them.

## E.3 STREAMING METRICS FOR TCUQ

Because **TCUQ** operates online, we complement static metrics with streaming criteria that capture drift response and budgeted abstention:

**Quantile risk control.** Let the nonconformity $r_t$ and maintained online quantile $q_{\alpha,t}$ be defined in Section 3. We track the exceedance deviation

$$\Delta_\alpha = \left| \frac{1}{T} \sum_{t=1}^{T} \mathbb{1}(r_t \geq q_{\alpha,t}) - \alpha \right|, \tag{11}$$

measuring how closely the calibrated rejection rate matches the target risk level $\alpha$ on ID segments (lower is better).

**Abstention budget adherence.** With a desired long-run budget $b$ and observed abstention rate $\hat{b}$, we report $|\hat{b} - b|$ over the full stream and short windows, assessing both average and burst compliance.

**Event detection under CID.** For CID streams, we evaluate accuracy-drop detection using the AUPRC with drop events defined from sliding-window accuracy, and we report median detection delay (in steps) from event onset (higher AUPRC and smaller delay are better).

**Failure detection and selective risk.** Following Xia & Bouganis (2023), we compute AU-ROC/AUPRC for distinguishing correct vs. incorrect predictions within ID/CID and for rejecting OOD. We also trace risk–coverage curves for selective prediction, where risk is the error rate among accepted samples and coverage is the non-abstained fraction.

