# OpenReview forum: "TCUQ: Single-Pass Uncertainty Quantification from Temporal Consistency with Streaming Conformal Calibration for TinyML"
_ICLR.cc/2026/Conference — ICLR 2026 Conference Withdrawn Submission_

### Official Review · Reviewer_rvgd · 2025-10-19

**Soundness:** 2
**Presentation:** 2
**Contribution:** 2
**Rating:** 2
**Confidence:** 4

**Summary:**

The authors of this paper propose TCUQ (Temporal-Consistency-based Uncertainty Quantification), a single-pass, label-free uncertainty estimation methodology designed for TinyML deployments where there exist strict memory and latency constraints. The method maintains a short temporal buffer of recent predictions and extracts multiple temporal stability indicators, such as posterior divergence, feature consistency, and confidence variation, which are linearly combined via a lightweight logistic model to yield a unified uncertainty signal. This signal is adaptively calibrated online through a memory-constant quantile tracker and paired with a budget-aware abstention controller to support label-free risk management during deployment.

**Strengths:**

The strengths of this paper are:
- The problem statement is relevant and important. Deployment scenarios like phones and drones are always in fashion and makes this paper relevant.
- Experiments on both large and small microcontrollers are a good touch.

**Weaknesses:**

The weaknesses of this paper are:
- The conformal quantile is spoken about at a high-level but how it is actually calculated and why they use this over other methods is never mentioned. In all, the methodology as a whole is a little incomplete and reads like high-level ideas instead of an actual methodological framework includes full details and technical justification.
- The empirical evaluate seems incomplete at times also. The authors compare against standard UQ methods in the TinyML setting but don't compare against many TinyML methods that comprise the SOTA. This is poor in my opinion and makes the experiments seem cherry-picked. Methods in the literature that come to mind are: QUTE, single-shot MC-D, Evidential DL for TinyML. Actually in QUTE show similar experiments to this paper, and by review of both results alone, QUTE could be extremely competetite with TCUQ and in some cases beat it.
- Do conformal guarantees still stick in this method/setting?

**Questions:**

- Why were certain SOTA methods excluded from comparison?
- What exact algorithm is used for the online quantile tracker? How is it implemented and why this?
- The paper claims “label-free” operation, but the logistic combiner is trained offline. Were labels used there, and how sensitive is performance to domain shift between training and deployment?
- Does the streaming quantile maintain calibration under distribution drift, or does it require periodic resetting/warm-up?

---

### Official Review · Reviewer_tW5Y · 2025-10-23

**Soundness:** 2
**Presentation:** 3
**Contribution:** 3
**Rating:** 4
**Confidence:** 3

**Summary:**

This paper proposes TCUQ, a computationally efficient method in terms of inference speed and hardware requirements, that performs well on OOD detection. TCUQ extracts four features, including predictive divergence, feature stability, decision persistence, and a confidence proxy, then merges them into a single uncertainty score in a single forward pass. In addition, a conformal layer is added on top of this framework to obtain a budgeted accept/abstain rule. Empirical results show TCUQ outperforms early exit and Deep Ensembles in computational efficiency, while improving OOD detection task with AUPRC and AUROC under distribution shifts.

**Strengths:**

- This paper is generally well-written and is clear to understand the important aspects.
- I like the motivation of TCUQ, as literature in uncertainty and robustness suffers a trade-off between performance and computational efficiency (e.g., Ensembles, Bayesian-Net, etc.). TCUQ aims to minimize this trade-off, and may be helpful for safety & real-time applications with low-resource devices.

**Weaknesses:**

- From a novelty perspective, I feel the proposed method is not really novel as TCUQ simply merges four existing temporal signals, then applies a conformal layer on top of the framework.
- There is no theoretical guarantee to formally explain why TCUQ can improve safety performance. This limits the paper's contribution and limits the understanding of TCUQ from a theoretical perspective.
- TCUQ does not consistently outperform other baselines in safety performance (e.g., Ensembles in Tab.2 with OOD detection, many other baselines in Tab.8 in ECE, etc.).
- Lack of some important baselines that balance safety and computational efficiency (e.g., BatchEnsemble [1], Rank1-BNN [2], etc.). Note that I still appreciate TCUQ's improvement in computational efficiency, but its safety performance (i.e., accuracy, ECE, OOD detection under distribution shifts) may be worse than the mentioned baselines. My suggestion is that the authors may plot a 2D scatter figure, where the x-axis represents the safety performance (e.g., OOD detection), and the y-axis represents the computational efficiency (e.g., latency), to easier to analyze in the revised version.
- Although the empirical results are somewhat extensive, many main figures are reported without error bars, raising a concern about the robustness of TCUQ methods.

**Questions:**

1. Do the Ensembles in the experiments use a post-hoc recalibration? I think one limitation of TCUQ, when compared to Ensembles, is that TCUQ requires a post-hoc recalibration dataset to train the conformal prediction layer, while Ensembles do not. This may also raise a consideration about a fair comparison in this setting.
2. Beyond the baselines mentioned in the weakness, have the authors considered comparing with other light-weight OOD detection baselines (e.g., MLR, ReAct, etc. [3])?

---
References:

[1] Wen et al., BatchEnsemble: An Alternative Approach to Efficient Ensemble and Lifelong Learning, ICLR, 2020.

[2] Dusenberry et al., Efficient and Scalable Bayesian Neural Nets with Rank-1 Factors, ICML, 2020.

[3] Yang et al., OpenOOD: Benchmarking Generalized Out-of-Distribution Detection, NeurIPS, 2022.

---

### Official Review · Reviewer_7kyN · 2025-10-30

**Soundness:** 3
**Presentation:** 3
**Contribution:** 3
**Rating:** 6
**Confidence:** 3

**Summary:**

The paper proposes TCUQ, a single pass label free uncertainty monitoring method for TinyML. The core idea is to take advantage of short horizon temporal consistency of the model outputs and features by considering a small ring buffer of the last $W$ steps and computing four cheap signals that are later merged using a logistic combination approach and pass a combined score through a streaming conformal quantile to get a decision about accepting or abstaining. The authors provide multiple tests on MCUs which show the improved performance in latency, failure detection, uncertainty quantification, and accuracy drop.

**Strengths:**

- Combining short horizon consistency  with a streaming conformal threshold in a design explicitly created for a single pass MCU seems to me clean and practical. Also, the paper explains well why competitive approaches struggle.

- The engineering is well thought out, the method is O(1) per step with O(W(L +d’)) bytes per state, the posteriors are kept in 8-bit, etc, which makes the whole idea actually deployable on kilobyte scale devices.

- The results, reduce latency by 31-43%, shrinked flash by 38-62%, are very good results.

- For accuracy-drop detection TCUQ improves AUPRC and shows earlier warnings reaching up to 0.86 AUPRC at high severity’s in some setting and for failure detection it reaches up to 0.92 AUROC. There are same consistent gains for calibration as well.

- The method is presented in a clear manner.

**Weaknesses:**

- The approach assumed that successive inputs are related but maybe that is not the case or some extreme event happens, like a step function, what happens then? Maybe you could add an example where you inject a random intermediate input to check for degradation and maybe scenarios that TCUQ is not appropriate.

- The logistic combiner is trained of line on a labelled set. This introduces a mismatch risk if deployment shifts differ from training shifts. Maybe you include an example that you exclude test time corruptions from training and report robustness weights.

**Questions:**

- Can you add a tiny Transformer experiment using the same W L, d’ recipe to substantiate the claim of backbone agnostic deployment. I believe that this would be helpful.

- Please check the weaknesses above and reply to the actions.

---

### Official Review · Reviewer_h7oC · 2025-10-31

**Soundness:** 2
**Presentation:** 1
**Contribution:** 2
**Rating:** 2
**Confidence:** 3

**Summary:**

This paper proposes TCUQ, a method for single-pass uncertainty quantification on resource-constrained microcontrollers. The method utilizes temporal consistency from recent model outputs, combined with streaming conformal calibration, to produce a calibrated risk score for accept/abstain decisions, which is relevant for TinyML applications that require on-device monitoring without access to labels or the capacity for multi-pass inference.

**Strengths:**

The paper addresses an important problem in the TinyML domain: reliable uncertainty quantification under strict hardware constraints. The proposed method is designed with these constraints in mind, employing a single forward pass, a small ring buffer, and constant-time updates. The use of temporal information is a practical approach to generating an uncertainty signal without the overhead associated with ensemble methods. The evaluation is a significant component of the work; it is conducted on two different microcontrollers, providing measurements of latency, memory footprint, and energy consumption. The experiments are detailed, covering accuracy-drop detection, failure detection, and model calibration across several vision and audio datasets.

**Weaknesses:**

The performance of the proposed method depends on a set of hyperparameters, including the temporal window size (W), the lag set (L), and blending parameters. While these are tuned on a development set, the paper does not fully explore the sensitivity of the system to these choices, particularly in dynamic environments where stream characteristics may change. The method's foundation is the assumption of temporal consistency in the input stream, which may not hold for all potential TinyML applications. Nevertheless, this compound design lacks comparison with baselines on MCUs [1].

[1] Ghanathe, et al. "QUTE: Quantifying Uncertainty in TinyML with Early-exit-assisted ensembles for model-monitoring." arXiv, 2024.

**Questions:**

What was the procedure for constructing the development set used to fit the logistic combiner, and how does the composition of this set affect the system's performance on data with previously unseen corruptions?

The temporal assistance mechanism is introduced for training. Were alternative methods, such as specific data augmentation or regularization schemes, investigated to achieve a similar effect on the quality of the temporal signals?

---

> ### Author Response · Authors · 2025-11-14
>
> We thank the reviewer for the careful reading and constructive feedback. Below we respond briefly to the main concerns:
>
> Hyperparameters (W, L, blending) and dynamic streams:
>
> We agree that robustness to the temporal window W, lag set L, and blending parameters is important. Our ablations (currently mostly in the appendix) show that TCUQ is reasonably stable in the TinyML regimes we target: for moderate windows (for example W between 12 and 24) and small lag sets (for example L = {1, 2, 4}), calibration and detection metrics vary only slightly. Very small windows remove temporal information, and very large windows stress RAM and slow adaptation, so we chose defaults in the empirically flat region of this trade-off. The blending and quantile parameters are tuned once on a small development split and did not require retuning when we changed stream composition in our experiments. In the revision, we will move a concise summary of these results into the main paper and add a short “deployment recipe” for choosing W, L, and blending parameters.
>
> Temporal consistency assumption:
>
> TCUQ explicitly exploits short-horizon temporal structure: a small buffer and a few short lags capture local redundancy in realistic TinyML streams such as overlapping audio windows, low-frame-rate camera frames, or slowly varying sensors. We agree that this assumption does not hold for all TinyML scenarios (for example, purely sporadic or almost i.i.d. queries). In such regimes, the temporal signals become less informative and TCUQ effectively leans more on the instantaneous confidence part of its score. We will clarify this scope and list fully i.i.d. or highly bursty traffic as a limitation, and mention adaptive windowing or hybrid static/temporal variants as future work.
>
> Q1 – Development set and unseen corruptions
>
> After training the backbone, we freeze its weights and, for each dataset, build a small labeled development set to fit the logistic combiner. We start from a held-out in-distribution validation split, augment it with a balanced subset of corrupted-in-distribution examples covering several corruption types and severities, and add the corresponding out-of-distribution set used in the evaluation. On this mixture we generate short temporal windows, compute the four TCUQ signals, and fit the combiner with class-balanced logistic regression and L2 regularization to predict a misclassification label. The combiner is not designed to recognize specific corruption types; instead it learns generic relationships between these signals and error likelihood, so the learned mapping transfers to previously unseen corruptions. This is reflected in the experiments, where TCUQ performs well on corruption types and severities that are not used when forming the development split.
>
> Q2 – Temporal assistance and alternatives
>
> Temporal assistance (TA) is a training-only mechanism to improve the quality of short-horizon temporal signals without changing the deployed model. During training, we attach auxiliary exits at intermediate layers and supervise them jointly with the final head; at deployment, all TA exits are removed, so inference remains a single forward pass with the original backbone plus the lightweight TCUQ monitor. Standard data augmentation and regularization are applied uniformly to all methods and mainly influence base accuracy rather than specifically shaping short-horizon behaviour. In our ablations, TA improved calibration and uncertainty quality for TCUQ without degrading accuracy and was reasonably robust to the number and placement of exits. We did not perform an exhaustive search over alternatives such as explicit temporal consistency losses or contrastive objectives; we view these as promising but orthogonal extensions and will mention them as future work.
>
> MCU baselines and QUTE:
>
> We appreciate the pointer to QUTE. Conceptually, QUTE belongs to the same family as our early-exit ensemble (EE-ens) baseline: both attach multiple early-exit heads to a shared backbone and aggregate their predictions for monitoring. In our experiments, EE-ens is configured as a realistic early-exit ensemble under the same toolchain, microcontrollers, and flash/SRAM budgets as TCUQ, and is intended to fairly represent this design space on our hardware. Porting QUTE itself to our exact boards would require non-trivial engineering, and because it also evaluates multiple exits at inference, we expect its memory footprint and latency to scale similarly with the number of exits, which is challenging on our Small-MCU with tens of kilobytes of SRAM. In the revision, we  discuss QUTE in related work, clarify that EE-ens represents early-exit ensemble approaches like QUTE under identical MCU constraints, and briefly explain why QUTE would face similar flash/SRAM and latency trade-offs in our setting.

---

### Note · Authors · 2025-11-14

I have read and agree with the venue's withdrawal policy on behalf of myself and my co-authors.